# Disruption of entire *Cables2* locus leads to embryonic lethality by diminished *Rps21* gene expression and enhanced p53 pathway

Tra Thi Huong Dinh[1,2,3,4†], Hiroyoshi Iseki[1,5†], Seiya Mizuno[1,4†], Saori Iijima-Mizuno[1,6], Yoko Tanimoto[1], Yoko Daitoku[1], Kanako Kato[1], Yuko Hamada[1], Ammar Shaker Hamed Hasan[1,7], Hayate Suzuki[1,7], Kazuya Murata[1,4], Masafumi Muratani[4,8], Masatsugu Ema[9,10], Jun-Dal Kim[11,12], Junji Ishida[11], Akiyoshi Fukamizu[11], Mitsuyasu Kato[4,13], Satoru Takahashi[1,4], Ken-ichi Yagami[1], Valerie Wilson[14], Ruth M Arkell[15], Fumihiro Sugiyama[1,4*]

[1]Laboratory Animal Resource Center, Faculty of Medicine, University of Tsukuba, Tsukuba, Japan; [2]Ph.D. Program in Human Biology, School of Integrative and Global Majors (SIGMA), University of Tsukuba, Tsukuba, Japan; [3]Department of Traditional Medicine, University of Medicine and Pharmacy, Ho Chi Minh City, Viet Nam; [4]Transborder Medical Research Center, Faculty of Medicine, University of Tsukuba, Tsukuba, Japan; [5]International Institute for Integrative Sleep Medicine (WPI-IIIS), University of Tsukuba, Tsukuba, Japan; [6]Experimental Animal Division, RIKEN BioResource Research Center, Tsukuba, Japan; [7]Doctor's Program in Biomedical Sciences, Graduate School of Comprehensive Human Science, University of Tsukuba, Tsukuba, Japan; [8]Department of Genome Biology, Faculty of Medicine, University of Tsukuba, Tsukuba, Japan; [9]Department of Stem Cells and Human Disease Models, Research Center for Animal Life Science, Shiga University of Medical Science, Otsu, Japan; [10]Institute for the Advanced Study of Human Biology (WPI-ASHBi), Kyoto University, Kyoto, Japan; [11]Life Science Center for Survival Dynamics, Tsukuba Advanced Research Alliance (TARA), University of Tsukuba, Tsukuba, Japan; [12]Division of Complex Bioscience Research, Department of Research and Development, Institute of National Medicine, University of Toyama, Toyama, Japan; [13]Department of Experimental Pathology, Faculty of. Medicine, University of Tsukuba, Tsukuba, Japan; [14]MRC Centre for Regenerative Medicine, School of Biological Sciences, SCRM Building, The University of Edinburgh, Edinburgh, United Kingdom; [15]John Curtin School of Medical Research, The Australian National University, Canberra, Australia

**\*For correspondence:**
bunbun@md.tsukuba.ac.jp

†These authors contributed equally to this work

**Competing interests:** The authors declare that no competing interests exist.

**Abstract** In vivo function of CDK5 and Abl enzyme substrate 2 (Cables2), belonging to the Cables protein family, is unknown. Here, we found that targeted disruption of the entire *Cables2* locus (*Cables2d*) caused growth retardation and enhanced apoptosis at the gastrulation stage and then induced embryonic lethality in mice. Comparative transcriptome analysis revealed disruption of *Cables2*, 50% down-regulation of *Rps21* abutting on the *Cables2* locus, and up-regulation of p53-target genes in *Cables2d* gastrulas. We further revealed the lethality phenotype in *Rps21*-deleted mice and unexpectedly, the exon 1-deleted *Cables2* mice survived. Interestingly, chimeric mice derived from *Cables2d* ESCs carrying exogenous *Cables2* and tetraploid wild-type embryo overcame gastrulation. These results suggest that the diminished expression of *Rps21* and the

completed lack of *Cables2* expression are intricately involved in the embryonic lethality via the p53 pathway. This study sheds light on the importance of *Cables2* locus in mouse embryonic development.

## Introduction

The mouse embryo at the blastocyst stage, consists of two layers: the outer trophectoderm and the inner cell mass (ICM). The ICM is characterized as pluripotent stem cells, from which the epiblast and primitive endoderm are derived (*Chazaud et al., 2006*; *Evans and Kaufman, 1981*; *Rossant et al., 2003*). Epiblast cells give rise to all cell types of the fetal tissues. The primitive endoderm produces the visceral endoderm (VE) extraembryonic yolk sac lining. Following implantation, Wnt, Nodal, and bone morphogenetic protein (BMP) signaling pathways are essential and coordinately control formation of the proximal–distal (P–D) axis during the egg cylinder stage and the subsequent conversion of this axis into the anterior–posterior (A–P) axis early in gastrulation (reviewed in *Arkell and Tam, 2012*; *Robertson, 2014*; *Shen, 2007*; *Ten Berge et al., 2008*; *Wang et al., 2012*; *Winnier et al., 1995*). Nodal and Wnt activity levels are dependent upon the BMP pathway interactions (*Robertson, 2014*; *Tam and Loebel, 2007*). The epiblast undergoes rapid cell proliferation and is sensitive to DNA damage, which may lead to p53-dependent checkpoint activation and result in apoptosis (*Heyer et al., 2000*; *Kojima et al., 2014*; *O'Farrell et al., 2004*). The primitive streak is formed by regional regulated expression of lineage-specific markers including *Brachyury* (*T*) and *Wnt3* via the Wnt/β-catenin pathway, to initiate the gastrulation stage. (*Rivera-Pérez and Magnuson, 2005*; *Tam et al., 2006*). While some murine axis and gastrulation signaling events are known, many other processes remain undiscovered.

Cdk5 and Abl enzyme substrate 1 (Cables1, also known as ik3-1) founded the Cables protein family, each member of which has a C-terminal cyclin box-like domain. Cables1 physically interacts with cyclin-dependent kinase 2 (Cdk2), Cdk3, Cdk5, and c-Abl molecules, and is phosphorylated by Cdk3, Cdk5, and c-Abl (*Matsuoka et al., 2000*; *Yamochi et al., 2001*; *Zukerberg et al., 2000*). Furthermore, in primary cortical neurons, c-abl phosphorylation of Cables1 augments tyrosine phosphorylation of Cdk5 to promote neurite outgrowth (*Zukerberg et al., 2000*). Cables1 also functions as a bridging factor linking Robo-associated Abl and the N-cadherin-associated β-catenin complex in chick neural retina cells (*Rhee et al., 2007*). Notably, *Cables1*-null mice show increased cellular proliferation resulting in endometrial hyperplasia, colon cancer, and oocyte development (*Kirley et al., 2005a*; *Lee et al., 2007*; *Zukerberg et al., 2004*). Additionally, the corpus callosum development in mice may rely on a dominantly acting, truncated version of Cables1 (*Mizuno et al., 2014*). During zebrafish development, *Cables1* is required for early neural differentiation and its loss subsequently causes apoptosis of brain tissue and behavioral abnormalities (*Groeneweg et al., 2011*). Zebrafish have only one Cables gene (Cables1), whereas the mouse and human genomes contain the paralogue, Cables2 (also known as ik3-2). The C-terminal cyclin-box-like region of Cables1 and Cables2 share a high degree of similarity. Cables2 has been shown to physically associate with Cdk3, Cdk5, and c-Abl (*Sato et al., 2002*). Moreover, forced expression of Cables2 induced apoptotic cell death in both a p53-dependent manner and a p53-independent manner in vitro (*Matsuoka et al., 2003*). In human, CABLES2 was recently found as a new susceptibility gene or tumor suppressor in colorectal cancer (*Guo et al., 2021*). Adult mouse tissues including the brain, testis, and ovary express Cables2 (*Hasan et al., 2021*), however, the role of this protein in vivo is unknown.

Therefore, in this study, we generated *Cables2d* mice with completely deleted entire locus (*Cables2d*) to elucidate Cables2 function in vivo. The data reveal the necessity of the *Cables2* locus for early embryonic development in mice and introduce the novel relationship of *Cables2* to *Rps21*, the p53 and Wnt/β-catenin pathways.

## Results

### Expression of *Cables2* during early mouse development

*Cables2* is widely expressed at equivalent levels in mouse tissues, including the brain, heart, muscle, thymus, spleen, kidney, liver, stomach, testis, skin, and lung (*Sato et al., 2002*). We first investigated

the expression of *Cables2* in mouse embryonic stem cells (ESCs), blastocysts, and embryos at E7.5 by reverse transcription polymerase chain reaction (RT-PCR). The results indicated that *Cables2* was expressed in all three stages of early development (*Figure 1A*). To confirm *Cables2* gene expression in mouse embryogenesis, localization of *Cables2* mRNA expression was examined in embryos by WISH (*Figure 1B-F*). The data for the whole embryo and transverse sections showed that *Cables2* was expressed at E6.5 (*Figure 1B,C*). *Cables2* was detected in both extra- and embryonic parts at E7.5 (*Figure 1D*) and strongly expressed in the allantois and in regions caudal to the heart at E8.5 (*Figure 1E*). At E9.5, widespread expression of *Cables2* was observed in embryo and extraembryonic tissues, including the yolk sac (*Figure 1F*). Overall, these data indicate that *Cables2* is widely expressed during early development, including throughout gastrulation in mouse embryos.

## Early embryonic lethality in *Cables2d* model

*Cables2d* mice were generated to investigate the physiological role of Cables2 in vivo. Cables2 heterozygous mice were produced using conventional aggregation with *Cables2*-targeted ESC clones purchased from KOMP (Knockout Mouse Programme). The entire *Cables2* allele was deleted with VelociGene KOMP design, a targeting strategy of IMPC (International Mouse Phenotyping Consortium) (*Figure 2A*). Interestingly, no homozygous *Cables2d* mice were observed following intercrossing heterozygous mice; however, the heterozygotes were viable and fertile (*Table 1*, *Figure 2—figure supplement 1A*). Embryos were collected and genotyped at various times during embryonic development to identify the critical point at which *Cables2* is essential for survival (*Table 1*). Homozygous *Cables2* mutant mice were detected in Mendelian ratios at E6.5–E9.5 but no homozygous embryos were observed at or beyond E12.5, indicating that *Cables2d* mice die and are resorbed between E9.5 and 12.5 (*Table 1*, *Figure 2—figure supplement 1B*). All the *Cables2d* embryos collected at E7.5–9.5 were considerably smaller than their wild-type littermates and did not progress beyond the wild-type early-mid-gastrula cylindrical morphology (*Table 1*, *Figure 2B–E*). Considerably small *Cables2^{d/d}* embryos had barely progressed beyond E8.5 (*Figure 2C*). Notably, at E7.5 homozygous mutant embryos resembled E6.5 wild-type embryos, in both morphology and size, when the primitive streak is just beginning to form (*Figure 2F,G*). Histological analyses confirmed that pre-streak stage (E6.0) *Cables2^{d/d}* embryos were structurally normal, exhibiting a normal-sized epiblast, extraembryonic ectoderm, and primitive endoderm (*Figure 2H,I*). These results suggested that *Cables2* full deletion causes growth and patterning arrest in gastrulation accompanied by post-gastrulation embryonic lethality.

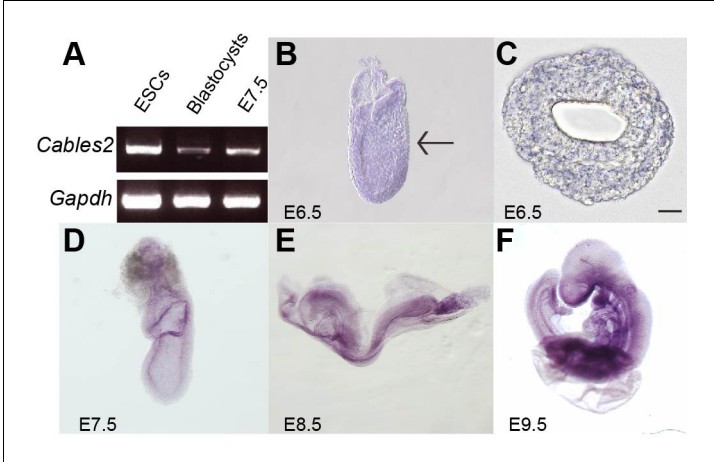

**Figure 1.** *Cables2* expression during early mouse embryo development. (**A**) *Cables2* gene expression was examined by RT-PCR with ESCs, blastocyst, and E7.5 embryo samples. *Gapdh* was used as an internal positive control. (**B–F**) Wild-type embryos from E6.5 to E9.5 were examined by in situ hybridization with a *Cables2* probe. The whole embryo expressed *Cables2* at E6.5 (**B**). The black arrow indicates the position of the transverse section shown in (**C**). Scale bars, 20 μm.

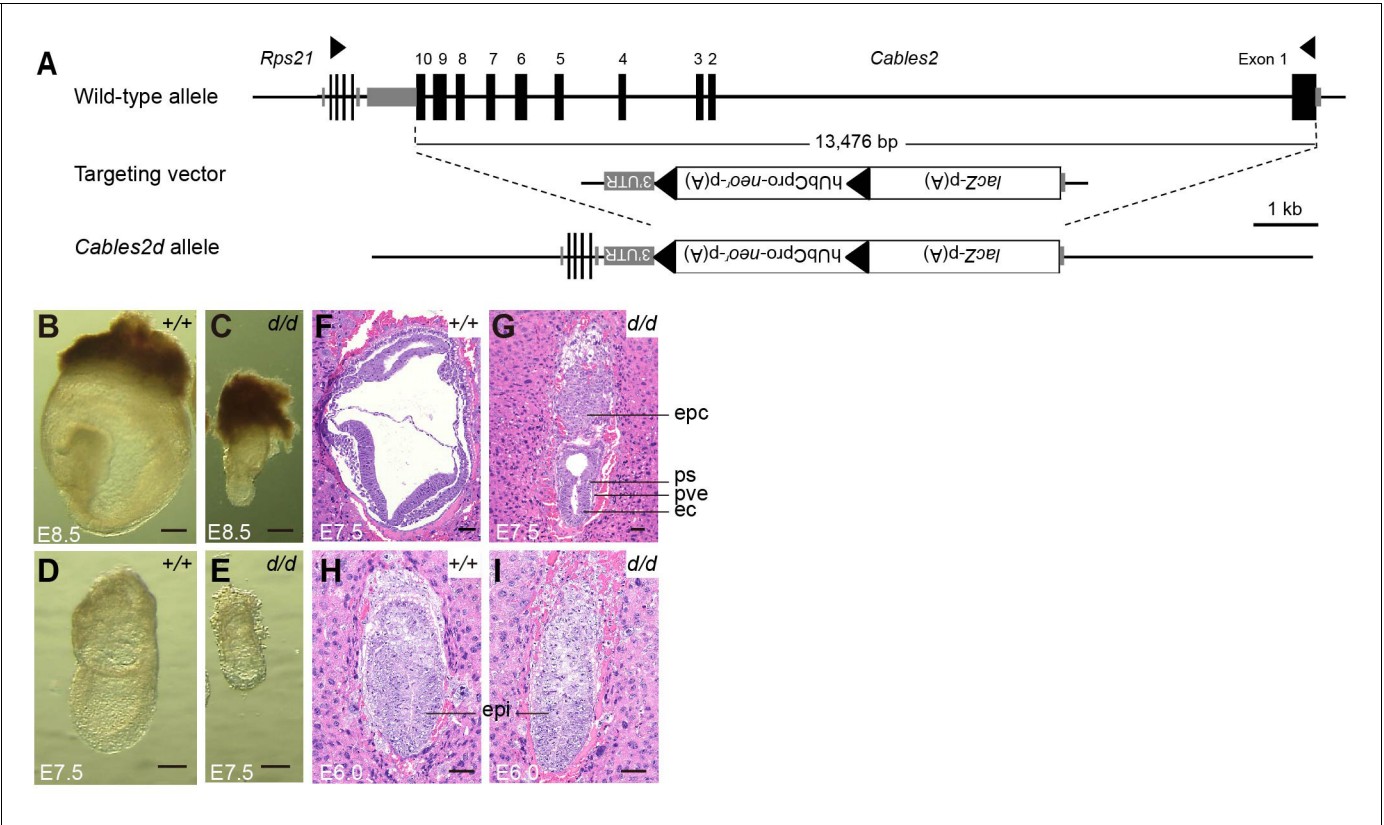

**Figure 2.** Morphological and histological analyses of *Cables2d* embryos at early stages of development. (**A**) Following VelociGene's KOMP design, the entire protein coding sequence of the target gene was deleted by homologous recombination in C57BL/6N ESCs, therefore, the full *Cables2* allele was null (*Cables2d* allele). Embryos were collected and genotyped at E8.5 (**B, C**) and E7.5 (**D, E**). Histological analysis was on HE-stained sections. Wild-type and *Cables2d* mutant embryos were embedded in paraffin and stained at E7.5 (**F, G**) and E6.0 (**H, I**). Epc: ectoplacental cone, ps: primitive streak; pve: posterior visceral endoderm; ec: ectoderm; epi: epiblast. Scale bars, 100 μm (**B–E**), 50 μm (**F–I**).

The online version of this article includes the following figure supplement(s) for figure 2:

**Figure supplement 1.** Genotyping of *Cables2d* and expression of wild-type *Cables2*.

## Developmental defect in *Cables2d* embryos at the onset of gastrulation

We further analyzed the expression of gastrulation markers at E6.5. Prior to gastrulation, *T* transcripts are first detected as a ring in extra-embryonic ectoderm and then in the posterior epiblast before the appearance of the primitive streak. (*Perea-Gomez et al., 2004*; *Rivera-Pérez and Magnuson, 2005*). *T* transcripts therefore serve as a marker of the transition from the P-D to A-P axis. At E6.5, *Cables2^{d/d}* embryos exhibited *T* expression as a band in the extra-embryonic ectoderm,

**Table 1.** Survival rate and Mendelian ratio of *Cables2*-mutant embryos.

| Embryonic days (E) | Total number of embryos | Genotypes | | |
|---|---|---|---|---|
| | | +/+ | +/- | -/- |
| E6.5 | 437 | 132 (30.2)* | 221 (50.6) | 80 (18.3) |
| E7.5 | 70 | 18 (25.7) | 32 (45.7) | 20† (28.6) |
| E8.5 | 21 | 9 (42.9) | 9 (42.9) | 3† (14.3) |
| E9.5 | 18 | 7 (38.9) | 7 (38.9) | 4† (22.2) |
| E12.5 | 6 | 2 (33.3) | 4 (66.7) | 0 (0) |
| Adult | 90 | 24 (26.7) | 66 (73.3) | 0 (0) |

* Number of embryos (percentage), † Abnormal phenotype.

indicating that the PS marker presented in the posterior epiblast even before morphological appearance of the primitive streak (*Figure 3A,B*). To confirm the PS formation in *Cables2*$^{d/d}$ embryos, we further investigated the expression of *Fgf8*, a member of the fibroblast growth factor family expressed in the PS (*Crossley and Martin, 1995*), and found that *Fgf8* was also appeared in mutant

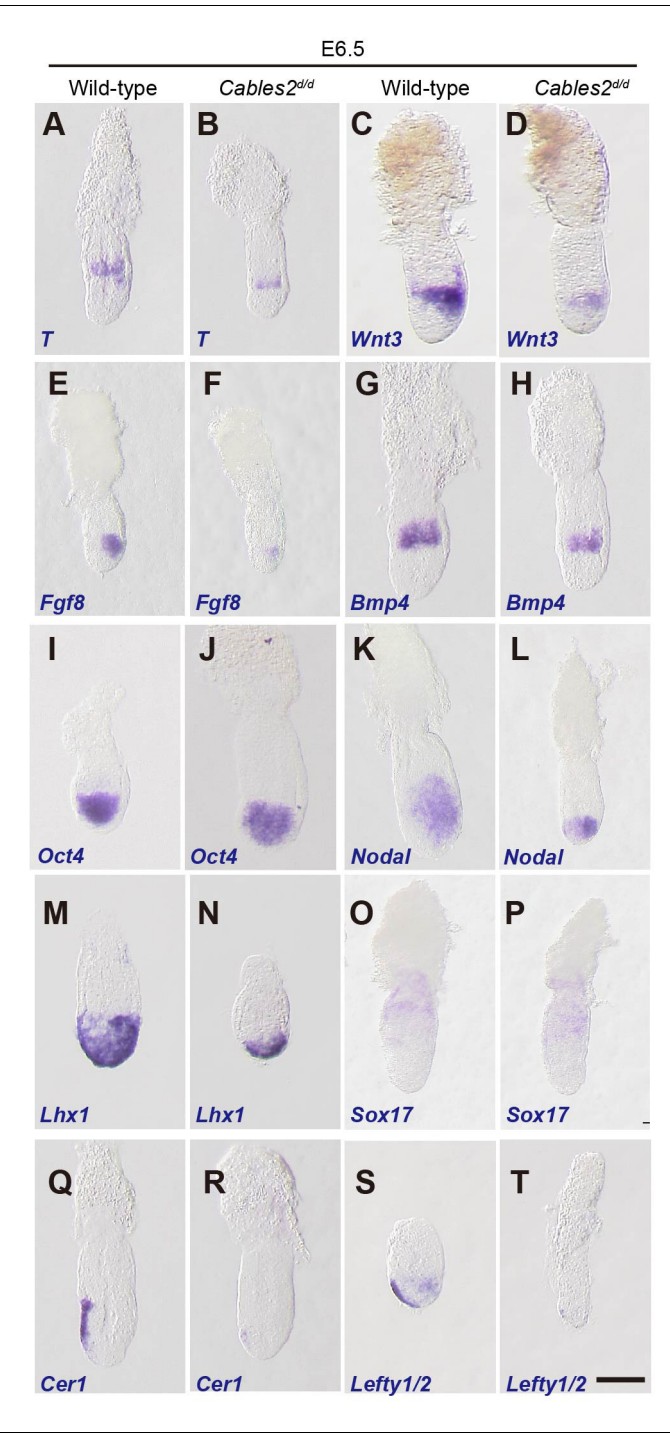

**Figure 3.** Expression of gastrulation markers in *Cables2d* embryos. (A–T) All embryos were collected, genotyped, and used for WISH at E6.5. Several key gastrulation markers were examined using both wild-type and *Cables2d* embryos: *T* (n = 5), *Wnt3* (n = 5), *Fgf8* (n = 3), *Bmp4* (n = 3), *Oct4* (n = 4), *Nodal* (n = 4), *Lhx1* (n = 3), *Sox17* (n = 3), *Cer1* (n = 3), and *Lefty1/2* (n = 3). Scale bars, 100 μm.

embryos at E6.5 (*Figure 3E,F*). Notably, *T* and *Fgf8* were decreased expression relative to wild-type embryos, suggesting the PS formation marker was initiated, however, with the low intensity in *Cables2$^{d/d}$* embryos.

PS formation and progression depends upon canonical Wnt signaling driven by the expression of *Wnt3* in the proximal-posterior epiblast and PVE (*Liu et al., 1999*; *Mohamed et al., 2004*; *Rivera-Pérez and Magnuson, 2005*; *Yoon et al., 2015*). *T* is a direct target of this Wnt activity (*Arnold et al., 2000*). WISH showed that expression of *Wnt3* appeared in the proximal and posterior part of E6.5 *Cables2$^{d/d}$* mutants with the decreased expression compared with wild-type (*Figure 3C, D*). In the extraembryonic part, *Bmp4* was similarly expressed in *Cables2$^{d/d}$* embryos compared with wild-type embryos at E6.5 (*Figure 3G,H*), suggesting that the extraembryonic ectoderm is develops normally in mutant embryos at least until E6.5. On the other hand, the pluripotency marker, *Oct4* was expressed normally in *Cables2$^{d/d}$* mutants at E6.5 (*Figure 3I,J*).

We next examined markers of the distal/anterior components of the axis in *Cables2$^{d/d}$* embryos. Proper activation of the Nodal signaling in VE is required for the AVE formation (*Takaoka and Hamada, 2012*). *Nodal* was normally expressed in *Cables2$^{d/d}$* embryos at E6.0 (data not shown). Subsequently, *Nodal* expression normally localizes to the nascent PS and the posterior epiblast at E6.5; however, in E6.5 *Cables2*-deficient embryos *Nodal* expression remains throughout the epiblast (*Figure 3K,L*). *Lhx1*, which is normally expressed in the AVE and nascent mesoderm of wild-type embryos, was accumulated in the distal part of E6.5 *Cables2d* embryos (*Figure 3M,N*). The normal formation of definitive endoderm and extraembryonic endoderm in mutant embryo was confirmed by the expression of *Sox17* (*Figure 3O,P*). Our data also showed that *Cerberus 1* (*Cer1*) and *Lefty1*, antagonists of Nodal signaling, were expressed at lower levels in *Cables2$^{d/d}$* embryos compared to the wild-type at E6.5 (*Figure 3Q–T*). Furthermore, WISH analyses demonstrated absent or decreased expression of *Lefty2* in the posterior epiblast of *Cables2d* embryos at E6.5 (*Figure 3S,T*). The combined results of WISH analyses suggested that PS formation is retarded in the *Cables2d* model but A-P axis is established.

## Activation and interaction of Cables2 with Wnt/β-catenin signaling

The Cables2 paralog (Cables1) binds to β-catenin (*Rhee et al., 2007*) and, in fact, the Wnt/β-catenin targets are downregulated in the *Cables2* mutant embryo (*Figure 3A–D*). We therefore examined whether Cables2 facilitates β-catenin activity at Wnt target sites and physically interacts with β-catenin. Cables2-activated β-catenin/TCF-mediated transcription in vitro with an almost twofold increase in relative TOP/FOP luciferase activity (*Figure 4A*). Moreover, co-IP using N-terminal FLAG-tagged Cables2 (FLAG-Cables2)-transfected 293T cell lysates with or without exogenous β-catenin indicated that β-catenin was present in the precipitated complexes with Cables2 (*Figure 4B* and *Figure 4— figure supplement 1*). These data suggested that Cables2 physically associates with β-catenin and increases its transcriptional activity at Wnt-responsive genes in vitro.

Accumulating evidence suggests that Wnt/β-catenin signaling is implicated in the formation of AVE and further PS (*Engert et al., 2013*; *Huelsken et al., 2000*; *Lickert et al., 2002*). To assess the functional significance of altered *Wnt3* ligand expression, *Cables2d* mice were crossed with the TOP-GAL transgenic mice, which express the β-galactosidase under the control of three copies of the Wnt-specific LEF/TCF binding sites (*Moriyama et al., 2007*). Beta-galactosidase was detected in the fully elongated primitive streak and in the adjacent posterior tissues as expected in wild-type E7.5 embryos carrying TOPGAL (*Figure 4C*). In contrast, E7.5 *Cables2$^{d/d}$* embryos carrying TOPGAL showed diminished β-galactosidase only in the proximal-posterior PS (*Figure 4D*). Moreover, *T* transcripts were observed in the PS of the mutant embryos, but not extending to the distal point of the embryo, and there was no signal in the axial mesendoderm (*Figure 4E,F*). These results suggested altered transcription activation of Wnt/β-catenin signaling in *Cables2$^{d/d}$* embryos.

## Increased apoptotic cells in *Cables2d* embryo at E7.5

Cell proliferation and apoptotic cell death are key events during embryonic development. To clarify the cell growth status, we performed EdU assay and measured the percentage of EdU-positive cells. There was no significant difference in the percentage of proliferation cells between wild-type and *Cables2$^{d/d}$* embryos at E6.5 (*Figure 5A,B,M*). Furthermore, a simultaneous TUNEL assay was performed to determine whether the reduced size of *Cables2$^{d/d}$* embryos could be attributed to

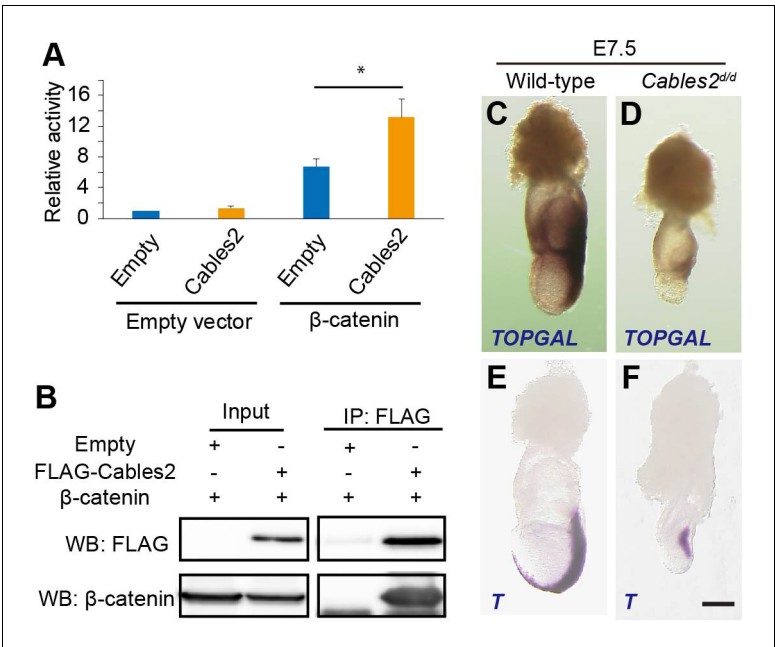

**Figure 4.** Enhancement of β-catenin activity by Cables2. (**A**) Relative luciferase activities in 293T cells transfected with an empty control or Cables2 expression vectors together with an empty control or β-catenin expression vectors. Relative luciferase activity is expressed as the ratio of TOP/FOPflash reporter activity relative to the activity in cells transfected with an empty vector alone. Columns: Averages of at least three independent experiments performed in triplicate. Error bars, Standard deviation (SD). Statistical significance was determined using Student's *t* test (*, p<0.05). (**B**) Co-IP was performed with FLAG-Cables2 and β-catenin expression vectors. The results obtained using anti-FLAG and anti-β-catenin antibodies showed the appearance of β-catenin in the precipitated complexes with Cables2. (**C, D**) β-Galactosidase staining demonstrating the restricted activation of Wnt/β-catenin signaling in *Cables2d* homozygous embryo carrying the TOPGAL reporter (*n* = 6). (**E, F**) WISH analysis showing the expression of *T* in wild-type and *Cables2d* embryos at E7.5 (*n* = 5). Scale bars, 100 μm.

The online version of this article includes the following figure supplement(s) for figure 4:

**Figure supplement 1.** Interaction of Cables2 with endogenous β-catenin in 293T cells.

increased programed cell death. Although apoptotic cells were detected, the average percentage of dead cells in *Cables2*$^{d/d}$ embryos was not significantly different from that in wild-type embryos (*Figure 5C–F,N*). These results suggest that cell proliferation and apoptotic cell death are normal in *Cables2d* embryos until E6.5. Interestingly, the percentage of proliferative cells of the wild-type and mutant embryos were comparable at E7.5 (*Figure 5G,H,M*), however, the TUNEL-positive apoptotic cells increased significantly in *Cables2*$^{d/d}$ E7.5 embryos while wild-type embryos exhibited few (*Figure 5I–L,N*). These results suggested that increased programed cell death occured in *Cables2*$^{d/d}$ model after E6.5.

## Decreased *Rps21* gene expression and elevated p53 pathway in *Cables2d* embryo

We performed RNA-seq with embryo samples at E6.5 and E7.5 to examine global gene changes. Kyoto Encyclopedia of Genes and Genomes (KEGG) pathways and Gene ontology (GO) terms were analyzed to find the maximum enrichment using the Enrichr program (*Chen et al., 2013*; *Kuleshov et al., 2016*). A heatmap resulted different expression genes in four groups with a cutoff of fold change > 2, FDR < 0.05 (*Figure 6A*, *Supplementary file 1*). At E6.5, the mutant embryos showed few differences from wild-type (fold change > 2, FDR < 0.05) with three genes downregulated and eight genes upregulated (*Figure 6B–D*, *Supplementary file 2*). Interestingly, aside from *Cables2*, *Rps12* and *Rps21* are significantly down-regulated in E6.5 mutant embryos (*Figure 6B*). *Rps12* was highly decreased at E6.5 in *Cables2d* embryos, however, not significantly different from wild-type embryos at E7.5. Importantly, *Rps21* is a gene located next to the exon 10 of the *Cables2*

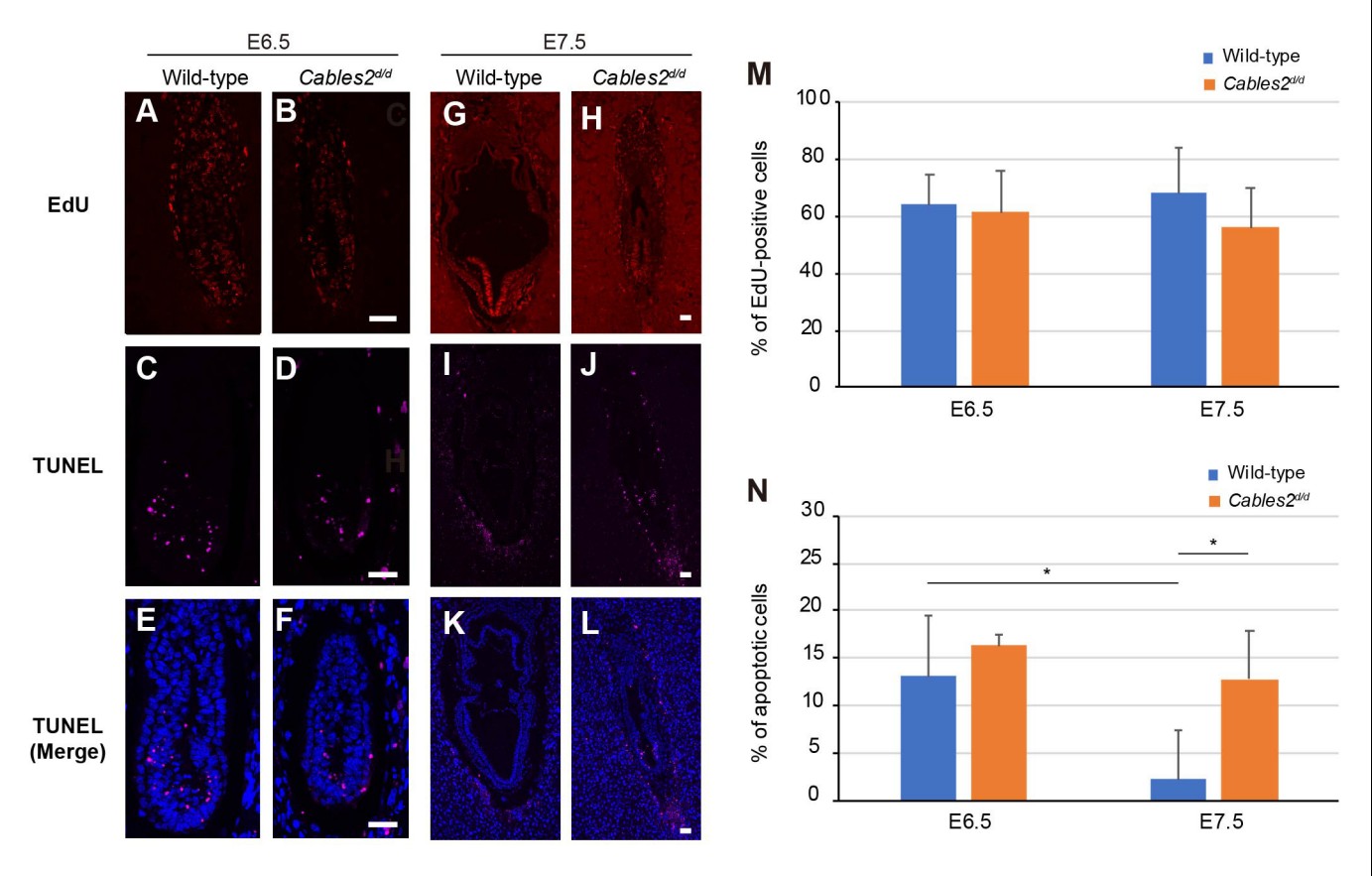

**Figure 5.** Proliferating and apoptotic cells in *Cables2d* embryos. (**A–B**) The EdU-incorporating cells represented the proliferation of cells in both wild-type and *Cables2*^d/d^ embryos at E6.5 (n = 6). (**C–F**) Apoptotic cells were detected in both embryonic and extraembryonic parts of wild-type and *Cables2*^d/d^ embryos (n = 6). (**G–N**) The proliferative and apoptotic cells at E7.5 were examined and showed the percentage in both wild-type and *Cables2*^d/d^ embryos (**M, N**). The average percentage was calculated by number of counted cells normalized to total number of cells within the embryo. Statistical significance was determined using two-way ANOVA (*, $p < 0.05$). Error bars, Standard deviation (SD). Scale bars, 50 µm.

locus in opposite orientation. We re-confirmed the quantitative decrease in *Rps21* mRNA in *Cables2*^d/d^ embryos at E7.5 (*Figure 6H*). Eight upregulated genes in *Cables2d* embryos including inducers and effectors of p53 indicated the 'p53 signaling pathway' is particularly enhanced without *Cables2* (*Figure 6C*). Enhanced p53 processes include cyclin-dependent protein kinase regulation and cell cycle arrest response to DNA damage (*Figure 6D*). The expression of significantly upregulated genes *Ccng1*, *Trp53inp1*, and *Cdkn1a (p21)* was quantitatively measured using E7.5 embryos. Notably, comparable *Trp53* expression suggests no impairment of *Trp53* transcription.

At E7.5, the differentially expressed genes of mutant *Cables2* and control embryos (fold change >2, FDR < 0.05) include 350 downregulated genes and 207 upregulated genes, indicating that the transcriptome signature was significantly disturbed at E7.5 (*Figure 6E*, *Supplementary file 3*). The enriched pathway of downregulated genes included 'axon guidance', 'Wnt signaling pathway', 'Hippo signaling pathway', 'PI3K-Akt signaling pathway' and the GO terms related to 'axonogenesis', 'nervous system development', 'circulatory system development' and 'heart development'. The upregulated KEGG pathways were 'mineral absorption', 'p53 signaling pathway' and the GO terms related to 'gonad development', 'male gonad development' and 'development of primary male sexual characteristics' (*Figure 6F,G*). These data revealed programmed cell death and the p53 pathway may impair gastrulation and contribute to *Cables2* mutant embryo lethality. Transcriptome profiling comparison suggested that disruption of entire *Cables2* locus affects the expression of not only *Cables2*, but also *Rps21*, gene abutting the *Cables2*, and induces the enhanced expression of p53-target genes dramatically.

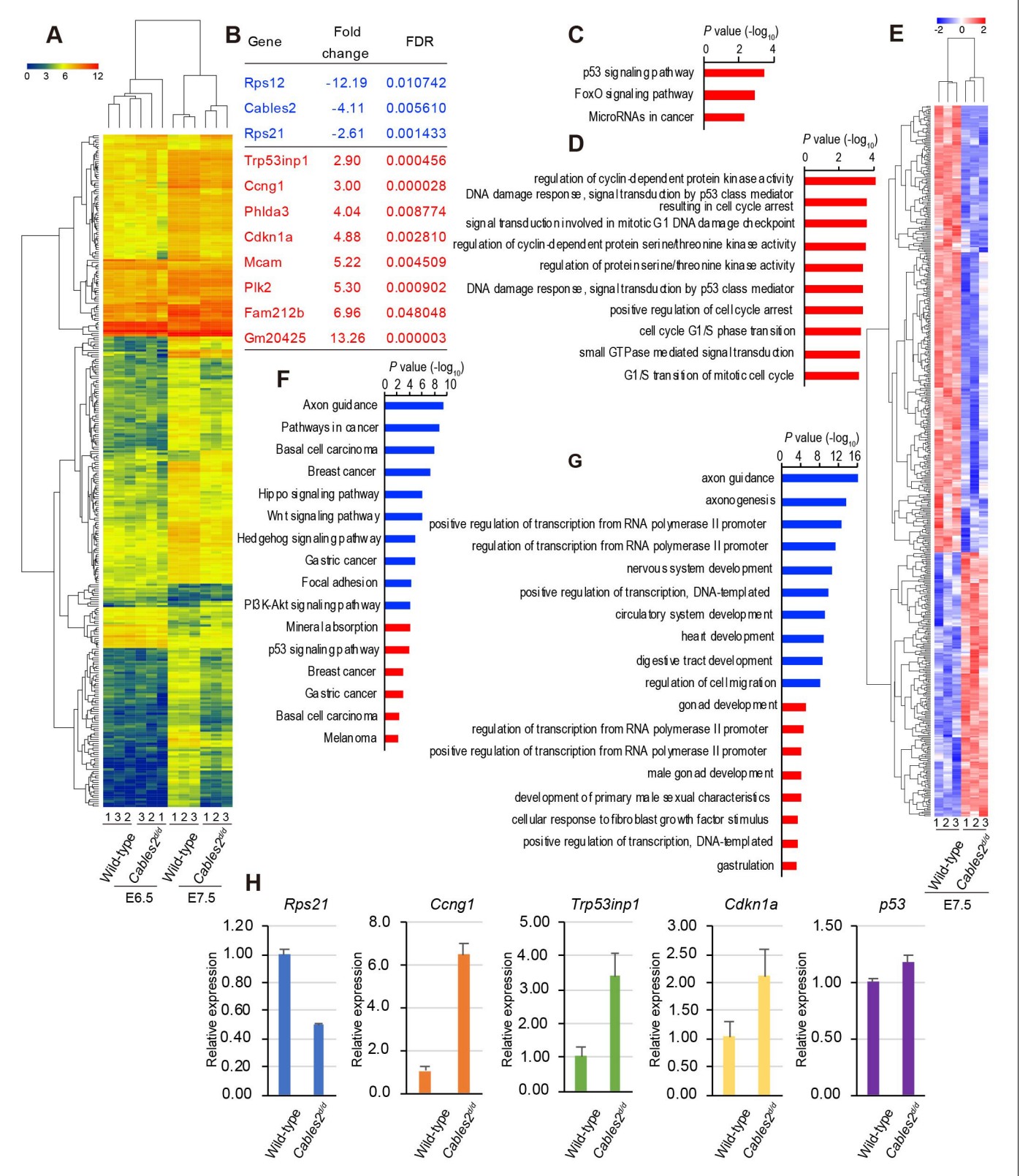

**Figure 6.** Transcriptome profiling analysis of *Cables2^{d/d}* embryos. (A) Heatmap representation of 288 genes significantly different between wild-type and *Cables2^{d/d}* embryo samples (fold change >= 2, FDR < 0.05). (B) List of all downregulated (blue) and upregulated (red) genes expressing in *Cables2d* embryos at E6.5. (C) KEGG pathway (p<0.01) and (D) GO Biological process (p<0.001) were identified among up-regulated genes at E6.5. (E)
*Figure 6 continued on next page*

*Figure 6 continued*

Heatmap (Z score) for the expression of 350 downregulated and 207 upregulated genes in *Cables2^{d/d}* embryos at E7.5. Different expression genes at E7.5 enriched in KEGG pathway (**F**) and GO Biological process (**G**) (p<0.001) with downregulation in blue bars and upregulation in red bars. (**H**) RT-qPCR validated the expression levels of representative upregulation genes at E6.5 using *Cables2^{d/d}* embryos E7.5 (n = 5). Averages of three independent experiments performed in duplicated and normalized against the expression levels of *Gapdh*. Error bars, Standard deviation (SD).

## The discrepant phenotypes in *Cables2* deletion models

The results of RNA-seq raise the question of which gene, *Cables2* or *Rps21*, is the main cause of the elevated expression of p53-target genes and embryonic lethality in *Cables2d* mice. To explore the specific function of *Cables2* in embryonic development, the *Cables2* conditional KO exon one mice was obtained using CRISPR/Cas9 system. During the modified-gene mouse production, the *Cables2* exon one deletion mice (*Cables2e1*) also were generated (**Figure 7A**). To re-confirm the lethal phenotype, *Cables2e1* mice were exclusively intercrossed and propagated. Unexpectedly, viable and fertile homozygous *Cables2^{e1/e1}* mice were obtained, which is contrary to entire locus *Cables2d* phenotype. This surprising result indicates an inconsistent function of *Cables2* in embryogenesis. RT-PCR analysis of adult mouse brain showed that expression of *Cables2* mRNA was deleted in *Cables2^{e1/e1}* tissues (data not shown). Moreover, the quantitative RT-PCR confirmed lack of *Cables2* expression in *Cables2^{e1/e1}* compared with *Cables2d* heterozygote and wild-type (**Figure 7B**), suggesting that *Cables2* is not transcribed in the *Cables2^{e1/e1}* mouse.

## Investigating the phenotype of *Rps21* deletion and *Rps21*-indel mice

We considered that *Rps21* may contribute to *Cables2d* lethality due to its significant downregulation. Rps21 is a factor for protein translation initiation and is a potential regulator for early differentiation of mammary epithelial stem cells using the HC11 mouse cell line (**Perotti et al., 2009**; **Török et al., 1999**). To date, to our knowledge, in vivo Rps21 function has not been reported in the

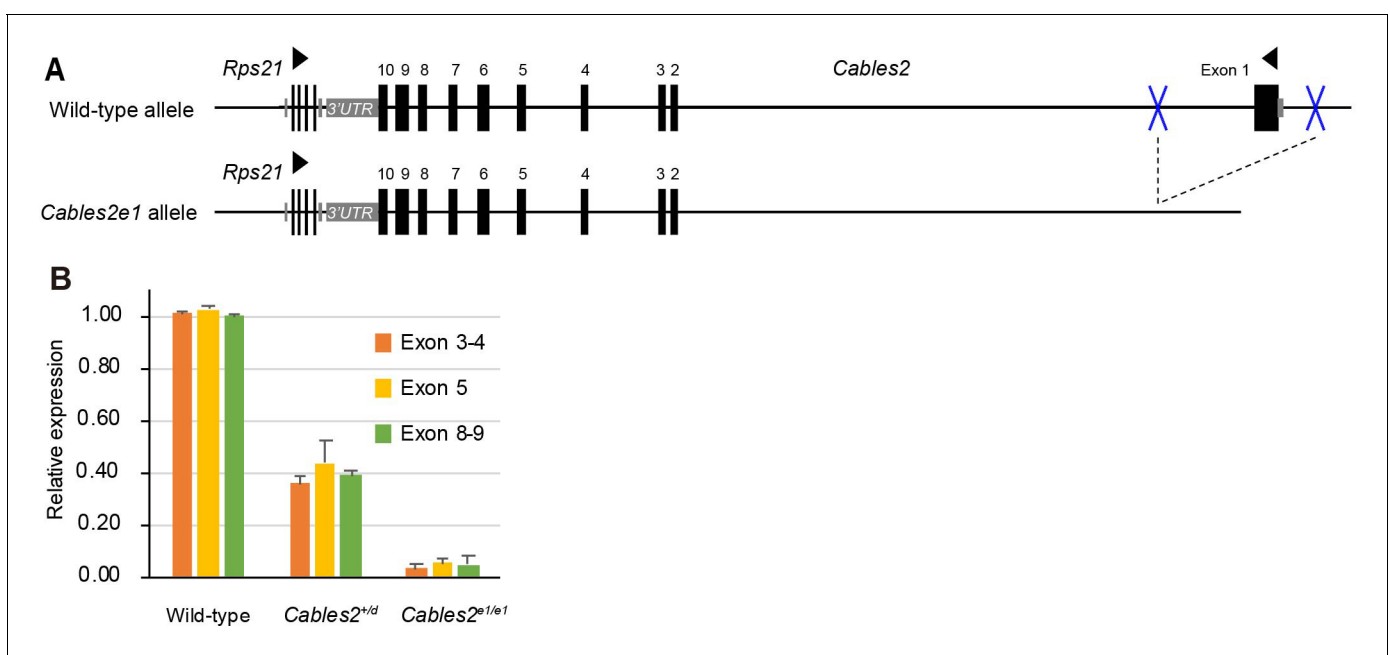

**Figure 7.** Gene construction and expression in *Cables2e1* mutant mouse. (**A**) The *Cables2e1* mice was generated using CRISPR/Cas9 system. Blue marks indicate the target sites on the left and right of exon1 of *Cables2*. (**B**) RT-qPCR using adult brain samples showed the quantitation of *Cables2* expression in wild-type, heterozygous *Cables2^{+/d}* and homozygous *Cables2^{e1/e1}* mice. The expression levels were validated in different exons of *Cables2*. Averages of three independent experiments performed in duplicated and normalized against the expression levels of *Gapdh*. Error bars, Standard deviation (SD).

The online version of this article includes the following figure supplement(s) for figure 7:

**Figure supplement 1.** Expression of compound embryos from *Cables2d* and *Cables2e1* intercrossing.

mouse model. To identify *Rps21* function, we generated *Rps21* deletion mice (*Rps21d*), in which exon 2 to exon 6 is deleted, and *Rps21*-indel mice with an identified frameshift occurs (*Figure 8*, *Supplementary file 4*). We could obtain 11 *Rps21d* heterozygous founders, but homozygous mice were not available (*Table 2*). Surprisingly, adult *Rps21⁺ᐟᵈ* mice exhibited white abdomen spots, white hind feet, and a kinked tail similar to *Bst* heterozygous mice, in which the *Rpl24* gene is mutated and homozygous mutant dies before E9.5 (*Oliver et al., 2004*). Notably, seven of eleven founders developed smaller body size compared with that littermates and died before 8 weeks. One remaining male founder was sterile, thus unable to produce progeny. The *Bst* phenotype was confirmed in *Rps21⁺ᐟᵈ* heterozygous F1 mice with extremely low fertility. On the other hand, no signs of pregnancy were observed in all eight *Rps21*-indel pseudopregnant mice unexpectedly. We performed Caesarean sections and no embryos were found, with only eight implantation sites detected in all the uteruses by 2% sodium hydroxide (data not shown) indicating prenatal lethality. These results suggested the semidominant phenotype of *Rps21* mutant models; however, the critical developmental stage regulated by *Rps21* remains unclear. Compared with *Cables2d* model, the heterozygote *Rps21d* showed more severe phenotype including small body size, infertility, and postnatal lethality or prenatal lethality in the case of *Rps21*-indel. These data indicated that *Rps21* could be essential for mouse development and the embryonic lethality in *Cables2d* mice resulted from diminished *Rps21* expression. However, it remains unclear why heterozygous *Rps21d* mice survived during the embryonic development, whereas *Cables2ᵈᐟᵈ* mice with 50% *Rps21* expression showed embryonic lethality.

## Overexpression of *Cables2* in the epiblast rescues the proper gastrulation stage in *Cables2d* tetraploid embryo

To determine function of *Cables2* in embryos with diminished *Rps21* expression, chimera analysis was performed using tetraploid wild-type embryos and *Cables2ᵈᐟᵈ* ESCs derived from the entire locus *Cables2d* embryos. Tetraploid complementation chimera have the advantage that the host tetraploid embryos can only contribute to primitive endoderm derivatives and trophoblast compartment of the placenta, whereas epiblast components are completely derived from ESCs (*Tanaka et al., 2009*). Tetraploid wild-type morula was aggregated with *Cables2ᵈᐟᵈ* ESCs to produce chimera in which *Cables2* was exclusively deleted in the epiblast (*Cables2d* Epi KO chimera) (*Figure 9A*). YFP reporter gene was inserted into *ROSA26* locus of *Cables2ᵈᐟᵈ* ESCs to construct *Cables2ᵈᐟᵈ*; *ROSA26ʸᶠᴾᐟ⁺* ESCs which allows for embryo visualisation and imaging. We collected the *Cables2* chimeric embryos at the indicated embryonic stage and analyzed the phenotype (*Table 3*). Like *Cables2ᵈᐟᵈ* embryos, the epiblast of *Cables2d* Epi KO embryos were smaller in size than that of control wild-type chimera littermates at E7.5 and E8.5 (*Figure 9B–E*). In similar to *Cables2ᵈᐟᵈ* gastrulas, growth retardation was observed in the *Cables2d* Epi KO chimera gastrulas. Next, tetraploid wild-type morula was aggregated with *Cables2ᵈᐟᵈ*; *ROSA26ʸᶠᴾᐟ⁺*; *CAG-tdTomato-2A-Cables2* ESCs to produce *Cables2d* Epi rescue chimera (*Figure 9F*). *CAG-tdTomato* fused *2A-Cables2* was randomly integrated into the genome of *Cables2ᵈᐟᵈ*;*ROSA26ʸᶠᴾᐟ⁺* ESCs to ubiquitously overexpress

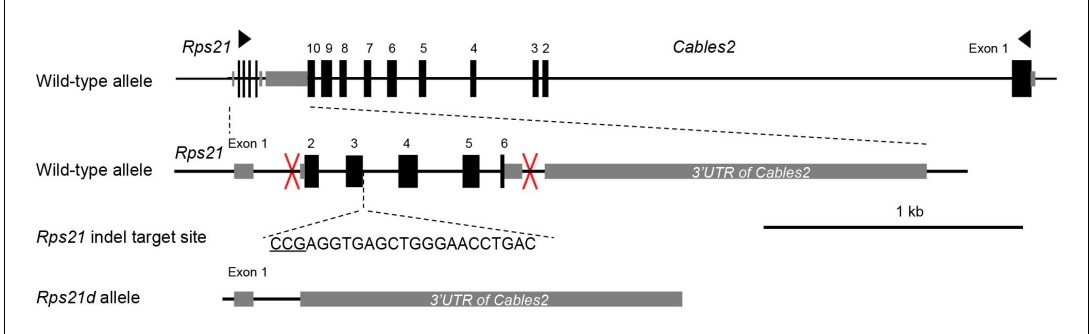

**Figure 8.** Gene construction of *Rps21d* and *Rps21*-indel mutant mice. *Rps21d* and *Rps21*-indel mutant mice were generated using CRISPR/Cas9 system. One-cut target site was designed to induce the mutation in *Rps21*-indel. Left and right target sites were introduce (red marks) to delete exon 2 to exon 6 of *Rps21* in *Rps21d* mouse. Schematic showed the full-length of *Rps21* including six exons and abutting to *3'UTR* of *Cables2* gene.

**Table 2.** Generation of *Rps21* mutant mice.

| Strain | Electroporated embryos | Transfer embryos | Total number of mice | | |
|---|---|---|---|---|---|
| | | | Newborn | Founder | |
| | | | | Male | Female |
| *Rps21d* | 207 | 199 | 38 | 7 | 4 |
| *Rps21*-indel | 274 | 200 | 0[*] | - | - |

* At E16.5, there were eight implanted sites.

Cables2 in the epiblast and its derivatives. Red fluorescence was detected in *Cables2^{d/d}*; *ROSA26^{YFP/+}*; *CAG-tdTomato-2A-Cables2* (*Cables2^{d/d}*; *ROSA26^{YFP/+}*; *tdT-C2*) ESCs, suggesting that tdTomato-2A-Cables2 was correctly translated in the cells (***Figure 9G–N***). Interestingly, *Cables2d* Epi rescue chimeras were indistinguishable from wild-type chimera littermates at all stages (***Figure 9G–N***). Altogether, the lethal phenotype of *Cables2^{d/d}* embryos was rescued by Cables2 exogenous overexpression until at least E9.5 during gastrulation.

## Discussion

In this study, we provided the first evidence regarding the physiological roles of Cables2 in mice. We demonstrated that *Cables2* is expressed widely during early embryonic development and full *Cables2* locus deletion caused defective PS formation, increased apoptotic cells, growth retardation, and post-gastrulation embryonic lethality. Many other mouse mutants with impaired A–P axis or PS formation either become highly dysmorphic or complete gastrulation without growth retardation but exhibit patterning defects. The divergent phenotype resulting from entire *Cables2* deletion suggests multiple faulty processes during early mouse development. Further, transcriptome profiling comparison showed *Cables2* and *Rps21* impairments enhanced Wnt and p53 signaling pathways, possibly contributing to peri-gastrulation arrest in entire locus *Cables2d* embryos. Notably, the heterozygous *Rps21^{+/d}* mice showed the semidominant *Bst* phenotype and almost died before two months old and no homozygous mutants were obtained, indicating embryonic lethality. The deletion of exclusive *Cables2* by removing critical exon one in *Cables2^{e1}* model resulted in live and fertile mice versus the lethal phenotype of entire locus *Cables2d* mice. However, the tetraploid complementation experiments demonstrated that the A–P axis was formed normally with the wild-type VE and trophoblast compartment, and the growth retardation in the epiblast can be rescued by overexpressing Cables2. Thus, the *Cables2/Rps21*-impaired genotype is lethal during gastrulation, and novel interaction of Cables2 with Wnt/β-catenin and p53 pathways can rescue mouse development.

Importantly, lack of *Cables2* transcription itself does not disrupt gastrulation in mice, as *Cables2^{e1/e1}* mice survive. Conventional full gene locus deletion may affect *Rps21* located in the proximity. Diminished expression of *Rps21* was observed when the entire *Cables2* locus was deleted. *Rps21* belongs to the ribosome family. In mammals, the ribosome family includes 79–80 ribosomal proteins and four ribosomal RNAs that play a vital role in ribosome biogenesis, a fundamental process for cellular proliferation, apoptosis, and maintenance (***Kressler et al., 2010***; ***Maguire and Zimmermann, 2001***). Ribosomal proteins function not only in protein synthesis but also in genetic diseases and tumorigenesis (***Lai and Xu, 2007***; ***Zhou et al., 2015***). Some ribosomal proteins are demonstrated in vivo as necessary factors for mouse development such as *Rplp1*, *Rpl24*, *Rpl38*, *Rps3*, and *Rps6* (***Kondrashov et al., 2011***; ***Oliver et al., 2004***; ***Panić et al., 2006***; ***Peng et al., 2019***; ***Perucho et al., 2014***). Recently, *Rps21* was described as a human oncogene, especially in prostate cancer and osteosarcoma (***Arthurs et al., 2017***; ***Liang et al., 2019***; ***Wang et al., 2020***). However, Rps21 in vivo function remains unknown. Our results revealed that the *Rps21* mutant model showed the semidominant phenotype of pleiotropic abnormality including postnatal lethality. This study is the first report of *Rps21d* mouse and describe the essential function of Rps21 for pre- and post-natal development in mice.

We found upregulation of p53 signaling pathway-related genes, such as *Ccng1*, *Trp53inp1* and *Cdkn1a*, in full locus *Cables2d* gastrulas that accompanied with decreased expression of *Rps21*. The transcription factor p53 is well-known to function in DNA damage responses and tumor suppression in cancer (***Vogelstein et al., 2000***; ***Zilfou and Lowe, 2009***). P53 activates checkpoint regulation in

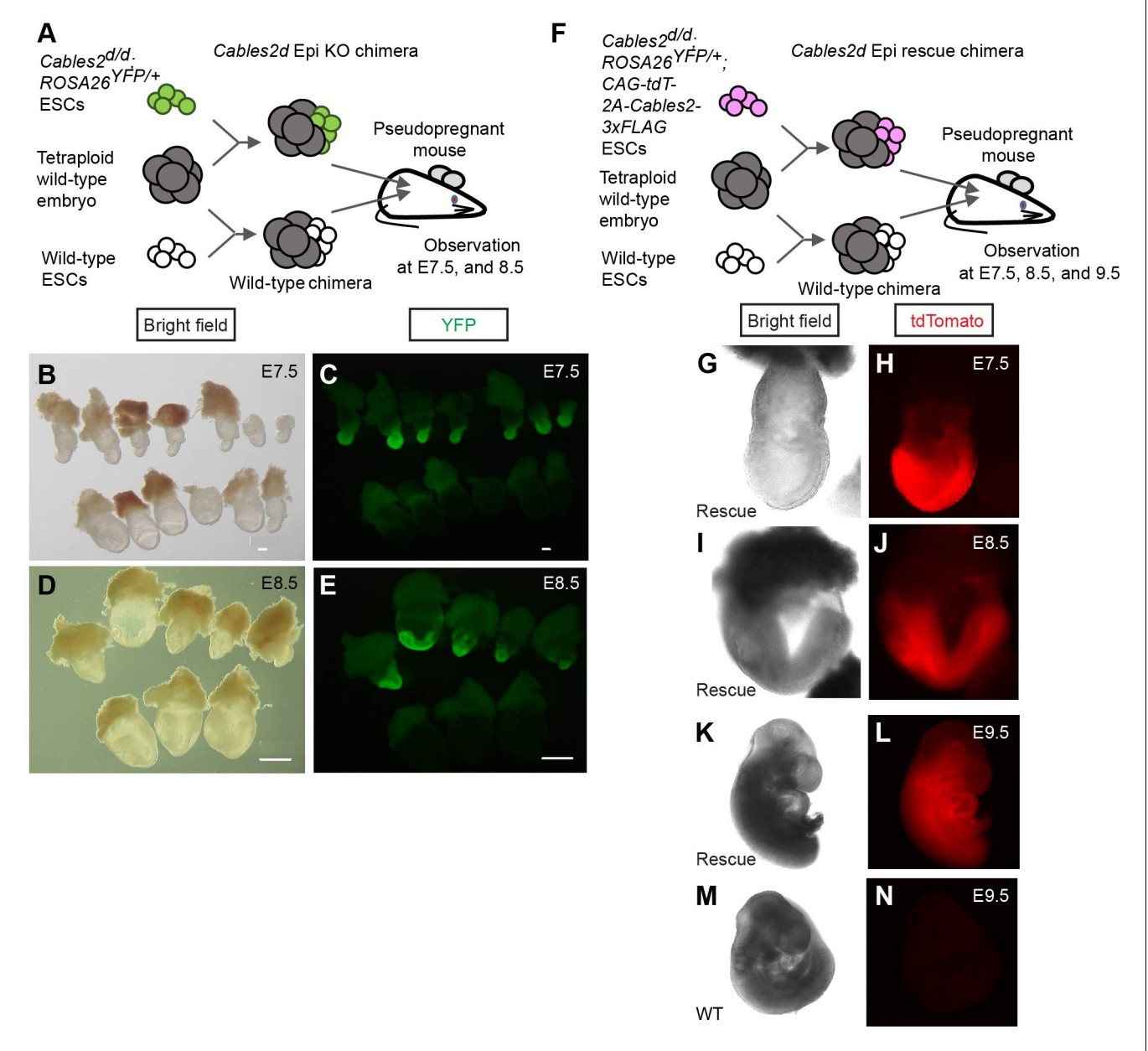

**Figure 9.** Defective and normal gastrulation development in *Cables2d* Epi KO and *Cables2d* Epi rescue chimeras, respectively. (A) Schematic diagram of tetraploid complementation experiment for *Cables2d* Epi KO chimera. (B–E) Bright field (B, D) and YFP fluorescent (C, E) images of wild-type and *Cables2d* Epi KO chimeric embryos at E7.5 or E8.5. (F) Schematic diagram of tetraploid complementation experiment for *Cables2d* Epi rescue chimeras. Bright field (G, I, K, M) and tdTomato fluorescent (H, J, L, N) images of wild-type and *Cables2d* Epi rescue chimeric embryos at E7.5, E8.5, or E9.5. The *Cables2d* Epi rescue chimeric embryos developed normally until E9.5. Scale bars, 100 μm (B, C); 500 μm (D, E).

ribosomal protein deficiency, rather than ribosome dysfunction. Interestingly, genetic deletion of p53 can rescue the lethal *Rps6* phenotype. The *Rps6* haploinsufficiency embryo exhibited peri-gastrulation lethality after E5.5 but can be rescued until E12.5 by genetic inactivation of p53 (Panic, et al., 2006). As mentioned before, heterozygous mutation in the *Rpl24* gene causes the *Bst* phenotype. Homozygous *Rpl24* deficiency shows early embryonic lethality. The abnormal postnatal phenotypes are largely caused by the aberrant up-regulation of p53 protein expression during embryonic development including the gastrulation stage (*Barkić et al., 2009*). These reports support that diminished expression of *Rps21* attenuates embryonic growth by reinforced induction of p53-dependent checkpoint response in entire locus *Cables2d* mice.

**Table 3.** Phenotypes in *Cables2d* Epi KO and *Cables2d* Epi rescue chimeras.

| Embryonic days (E) | Tetraploid embryo + Cables2$^{d/d}$; ROSA26$^{YFP/+}$ ESC (Epi KO chimera) | | | Wild-type chimera | | |
|---|---|---|---|---|---|---|
| | Total number of embryos | Phenotype Normal | Phenotype Abnormal | Total number of embryos | Phenotype Normal | Phenotype Abnormal |
| E6.5 | 2 | 1 | 1 | 4 | 3 | 1 |
| E7.5 | 15 | 4 | 11 | 14 | 11 | 3 |
| E8.5 | 13 | 0 | 13* | 7 | 6 | 1 |

| Embryonic days (E) | Tetraploid embryo + Cables2$^{d/d}$; ROSA26$^{YFP/+}$; CAG-tdTomato-2A-Cables2-3xFLAG ESC (Epi rescue chimera) | | | Wild-type chimera | | |
|---|---|---|---|---|---|---|
| | Total number of embryos | Phenotype Normal | Phenotype Abnormal | Total number of embryos | Phenotype Normal | Phenotype Abnormal |
| E7.5 | 5 | 4 | 1 | 2 | 1 | 1 |
| E8.5 | 4 | 3 | 1 | 5 | 4 | 1 |
| E9.5 | 2 | 2 | 0 | 3 | 2 | 1 |

* All embryos had A-P axis specification.

Heterozygous deficiency of *Rps21* showed the *Bst* phenotype similar to that of *Rpl24*, but not that of *Rps6*. Interestingly, although expression of *Rps21* was maintained at a half level in *Cables2d* gastrulas compared with wild-type gastrulas, they caused early embryonic lethality that was different abnormality from heterozygous *Rps21d* mice. Further, we generated *Cable2$^{d/e1}$* mutant embryos by intercrossing between *Cables2$^{+/d}$* and *Cables2$^{+/e1}$*. The compound *Cable2$^{d/e1}$* embryos showed normal development by E9.5 with Mendelian ratio (**Supplementary file 6**) and maintained at approximately 75% on expression levels of *Rps21*, suggesting that a threshold level of *Rps21* expression is required for early mouse embryonic development (**Figure 7—figure supplement 1**). However, it is thought that phenotype discrepancy between *Cables2$^{d/d}$* and *Rps21$^{+/d}$* would be attributed to the complete loss of Cables2 function, in addition to the hypofunction of Rps21. Actually, the impaired *Rps21/Cables2* epiblast can be rescued and developed to the early organogenesis period through the gastrulation stage by inducting ubiquitous overexpression of exogenous *Cables2* throughout embryonic development. Therefore, our results imply that Cables2 itself could play the role of fetal growth by resisting activation of p53 signaling pathway which results from a decrease in Rps21.

Genetic experiments that ablate *Wnt3* activity in either the VE or epiblast alone (and therefore reduce overall posterior Wnt signal) have shown that this activity regulates the timing of PS formation. Embryos in which VE *Wnt3* function is ablated have delayed primitive streak formation but by E9.5 are indistinguishable from wild-type littermates (**Yoon et al., 2015**). When *Wnt3* function is removed from the epiblast, PS formation is delayed and by mid-gastrulation the embryos are highly dysmorphic with the PS bulging toward the amniotic cavity (**Tortelote et al., 2013**). In entire locus *Cables2d* embryos, the WISH analysis with gastrulation markers showed that PS formation is retarded or delayed at gastrulation initiation with the reducing of *Wnt3* expression and activity. However, the cellular apoptosis was increased afterward and caused the growth failure. In **Figure 5O**, TUNEL-positive cells were abundantly observed in the epiblast of *Cables2d* embryos at E7.5, but not in the visceral endodermal cells. The result suggests that diminished Wnt/β-catenin signaling with *Cables2* deficiency might be involved in retardation of AVE and/or PS formation rather than diminished level of Rps21. In addition, Rps21 expression is not reported in embryo endoderm and mesoderm by MGI database (http://www.informatics.jax.org), further supporting our hypothesis.

Cables1 is widely studied in cancer research. It is generally thought that *Cables1* is a tumor suppressor gene. Loss of Cables1 expression is found with high frequency in human cancer such as lung, colon, ovarian, and endometrial cancers (**Arnason et al., 2013**; **Kirley et al., 2005a**; **Kirley et al., 2005b**; **Zukerberg et al., 2004**). In mouse model, genetic inactivation of *Cables1* leads to an increased incident of endometrial cancer (**Zukerberg et al., 2004**) and colon cancer (**Arnason et al., 2013**; **Kirley et al., 2005a**). *Cables1*-deficient mouse embryonic fibroblast cells exhibit an increased

growth rate (*Kirley et al., 2005b*). Further, Cables2 shows involvement of in both p53-dependent and p53-independent apoptotic pathway by using in vitro analyses (*Matsuoka et al., 2003*). Given that Cables family share the identity of amino acid more than 70% at C-ter (*Sato et al., 2002*), although we expected that Cables2 may have similar function with Cables1 in mouse and human, the present study provided us controversial observation on the physiological role of Cables2. Recently, *Shi et al., 2015* reported that Akt (Ser/Thr kinase)-phosphorylation-mediated 14-3-3 binding prevents the apoptosis-inducing function of Cables1 for cell growth. Emerging evidence reveals that Cables1 can interact with a variety of proteins (*Arnason et al., 2013*; *Kirley et al., 2005a*; *Kirley et al., 2005b*; *Zukerberg et al., 2004*). Cables1 changes physiological phase dependent on counter partner protein. This study also demonstrated that Cables2 physically interacted with β-catenin. Moreover, we are understanding that Cables2 has the ability to physically interact with one kind of self renewal/pluripotent factor in vitro (data not shown). Therefore, it seems that Cables family protein is a signaling hub for the regulation of the cell cycle, cell growth, cell death and differentiation. However, how Cables2 controls temporal, spatial and physical interaction in vivo for resisting p53 signaling pathway during gastrulation remains unknown and needs further investigation.

Lastly, this study highlights the need to validate target knock-out genes as well as nearby genes in lethal phenotypes. In conclusion, our results suggest that *Rps21* expression is essential for gastrulation and *Cables2* assists it in case of decreased ribosomal biogenesis, via an unknown mechanism. Furthermore, Cables2 functions together with Wnt/β-catenin and p53 pathways in early embryonic development. These novel interactions should be expanded in future studies to give insight into the function of the Cables protein family and uncover additional roles for this protein.

## Materials and methods

### Key resources table

| Reagent type (species) or resource | Designation | Source or reference | Identifiers | Additional information |
|---|---|---|---|---|
| Gene (*M. musculus*) | Cables2 | PMID:11955625 | MGI:2182335 | |
| Gene (*M. musculus*) | Rps21 | PMID:10022917 | MGI:1913731 | |
| Strain, strain background (*M. musculus*) | B6.Cg-Tg(TOPGAL) | Riken BRC | RBRC05918 | |
| Cell line (*M. musculus*) | Cables2[tm1(KOMP)Vlcg] | KOMP | ID: VG16085, clone: 16085A-D3 RRID:MMRRC_052978-UCD | ESC line |
| Cell line (*Homo-sapiens*) | 293T | ATCC | CRL-3216 RRID:CVCL_0063 | embryonic kidney |
| Antibody | Anti-FLAG (M2) (mouse monoclonal) | Sigma-Aldrich | Cat# F1804, RRID:AB_262044 | WB(1:1000), IP (1:650) |
| Antibody | Anti-β-catenin (D10A8) (rabbit monoclonal) | Cell Signaling Technology | Cat# 8480, RRID:AB_11127855 | WB(1:1000) |
| Antibody | Anti-GAPDH (rabbit polyclonal) | Santa Cruz Biotechnology | Cat# sc-25778, RRID:AB_10167668 | WB(1:1000) |
| Antibody | Normal mouse IgG (mouse isotype control) | Santa Cruz Biotechnology | Cat# sc-2025, RRID:AB_737182 | IP(1:250) |
| Antibody | Anti-Mouse IgG-HRP (secondary antibody) | GE healthcare | Cat# NA931, RRID:AB_772210 | WB(5000) |
| Antibody | Anti-Rabbit IgG-HRP (secondary antibody) | GE healthcare | Cat# NA934, RRID:AB_772206 | WB(5000) |
| Peptide, recombinant protein | TrueCut Cas9 Protein | Thermo Fisher Scientific | Cat# A36498 | |
| Peptide, recombinant protein | Dynabeads Protein G | Thermo Fisher Scientific | Cat# 10003D | |

*Continued on next page*

*Continued*

| Reagent type (species) or resource | Designation | Source or reference | Identifiers | Additional information |
|---|---|---|---|---|
| Commercial assay or kit | GeneArt Precision gRNA Systhesis Kit | Thermo Fisher Scientific | Cat# A29377 | |
| Commercial assay or kit | Lipofectamine 3000 Reagent | Thermo Fisher Scientific | Cat# L3000015 | |
| Commercial assay or kit | AmpliTaq Gold 360 Master Mix | Thermo Fisher Scientific | Cat# 4398886 | |
| Commercial assay or kit | RNeasy Mini Kit | Qiagen | Cat# 74104 | |
| Commercial assay or kit | TB Green Premix Ex Taq II | Takara | Cat# RR820B | |
| Commercial assay or kit | Dual-Glo Luciferase assay system | Promega | Cat# E2920 | |
| Commercial assay or kit | Click-iT Plus EdU Imaging Kit | Thermo Fisher Scientific | Cat# C10638 | |
| Commercial assay or kit | Click-iT Plus TUNEL Assay for In situ Apoptosis Detection kit | Thermo Fisher Scientific | Cat# C10619 | |
| Software | CLC Genomics Workbench | Qiagen | RRID:SCR_011853 | |

## Animals and husbandry

ICR mice were purchased from CLEA Japan Co. Ltd. (Tokyo, Japan); C57BL/6N mice were purchased from Charles River Laboratory Japan Co. Ltd (Yokohama, Japan). For production of staged embryos, the day of fertilization as defined by the appearance of a vaginal plug was considered to be embryonic day 0.5 (E0.5). Animals were kept in plastic cages (4–5 mice per cage) under specific pathogen-free conditions in a room maintained at 23.5 ± 2.5°C and 52.5 ± 12.5% relative humidity under a 14 hr light:10 hr dark cycle. Mice had free access to commercial chow (MF; Oriental Yeast Co. Ltd., Tokyo, Japan) and filtered water throughout the study. Animal experiments were carried out in a humane manner with approval from the Institutional Animal Experiment Committee of the University of Tsukuba (#19–057 and #20–065) in accordance with the Regulations for Animal Experiments of the University of Tsukuba and Fundamental Guidelines for Proper Conduct of Animal Experiment and Related Activities in Academic Research Institutions under the jurisdiction of the Ministry of Education, Culture, Sports, Science, and Technology of Japan.

## Generation and genotyping of target gene-deficient mice

The targeted ESC clone *Cables2*^tm1(KOMP)Vlcg was purchased from KOMP (project ID: VG16085, clone number: 16085A-D3, RRID:MMRRC_052978-UCD). To generate *Cables2d* mice, ESCs were aggregated with the wild-type morula and transferred to pseudopregnant female mice. Male chimeras that transmitted the mutant allele to the germ line were mated with wild-type females to produce *Cables2d* mice with the C57BL/6N background. To produce *Cables2* conditional KO exon 1, *Cables2* exon 1 CRISPR left and right target sites were designed (*Supplementary file 4*), oligo DNAs were annealed, purified, inserted into the *pX330* vector (Addgene plasmid 42230, a gift from Dr. Feng Zhang [*Cong et al., 2013*]) and checked the in vitro activity by EGxxFP system (*Fujihara and Ikawa, 2014*). The ssDNA donors were co-injected with CRISPR vector to insert loxP and *EcoRI* to upstream, loxP and *EcoRV* to downstream of exon 1, respectively. Then *Cables2* exon one deletion mice exclusively were propagated and analyzed.

*Rps21d* and *Rps21*-indel mutant mice were generated using CRISPR/Cas9 system. Firstly, gRNAs were synthesised using GeneArt Precision gRNA Systhesis Kit (Thermo Fisher Scientific). Then the sgRNAs were co-electroporated with Cas9 protein using TrueCut Cas9 Protein v2 (Thermo Fisher Scientific) into mouse zygote (*Supplementary file 4*).

In all strains, adult mice were genotyped using genomic DNA extracted from the tail. For whole-mount in situ hybridization, embryos were genotyped using a fragment of yolk sac and Reichert membrane. Samples were dispensed into lysis solution (50 mM Tris-HCl, pH 8.5, 1 mM EDTA, 0.5%

Tween 20) and digested with proteinase K (1 mg/mL) at 55℃ for 2 hr, inactivated at 95℃ for 5 min, and then subjected to PCR. For paraffin slides, embryos were genotyped using tissue picked from sections and digested directly with proteinase K (2 mg/mL) in PBS. For others experiments, after collecting data, the whole embryos were used for genotyping. Genotyping PCR was performed with AmpliTag Gold 360 Master Mix (Thermo Fisher Scientific, Tokyo, Japan) and primer listed in *Supplementary file 6*.

## TOPGAL reporter mice

B6.Cg-Tg(TOPGAL) transgenic mice carrying LEF/TCF reporter of Wnt/β-catenin signaling were used for visualizing Wnt signaling pathway in vivo. TOPGAL mice were obtained from Riken BRC (RBRC02228). Animals were kept and maintained under the same conditions as described above. To produce the TOPGAL reporter in the homozygous *Cables2* background, TOPGAL heterozygotes were crossed with *Cables2* heterozygotes subsequently and finally, homozygous *Cables2* carrying TOPGAL transgene were collected at E7.5 together with littermates. All embryos were stained S-gal (*Sundararajan et al., 2012*) and then genotyped using both *Cables2* genotyping primers and TOP-GAL primers (*Supplementary file 6*).

## Cell culture

293T cells were obtained from The American Type Culture Collection (Manassas, Virginia). Cells were cultured in Dulbecco's modified Eagle's medium (DMEM) supplemented with 10% heat-inactivated fetal bovine serum. Mouse embryonic stem cells (ESCs) were maintained on 0.1% gelatine-coated dishes in mouse ESC medium consisting of DMEM containing 20% knockout serum replacement (KSR; Thermo Fisher Scientific), 1% non-essential amino acids (Thermo Fisher Scientific), 1% GlutaMAX (Thermo Fisher Scientific), 0.1 mM 2-mercaptoethanol (Thermo Fisher Scientific), and leukemia inhibitory factor (LIF)-containing conditioned medium, supplemented with two chemical inhibitors (2i), that is 3 μM CHIR99021 (Stemgent inc, Cambridge, Massachusetts) and 1 μM PD0325901 (Stemgent).

## RT-PCR, RT-qPCR, and RNA-seq

Cultured ESCs, about 130 blastocysts, and 21 embryos at E7.5 were collected. Total RNAs from blastocysts and embryos were extracted using Isogen (Nippon Gene Co., Ltd., Tokyo, Japan). RNA from ESCs was collected using an RNeasy Mini Kit (Qiagen K.K., Tokyo, Japan). The cDNA was synthesized using Oligo-dT primer (Thermo Fisher Scientific) and SuperScript III Reverse Transcriptase (Thermo Fisher Scientific) in a 20 μL reaction mixture. RT-qPCR was performed using TB Green Premix Ex Taq II (Takara) and the Thermal Cycler Dice Real Time System (Takara) according to the manufacturer's instructions and target gene expression level was normalized to the endogenous *Gapdh* expression level (*Supplementary file 6*).

RNA sequencing analysis was performed by Tsukuba i-Laboratory LLP as previously described (*Ohkuro et al., 2018*). Briefly, total RNAs were extracted from wild-type and *Cables2d* embryos, two embryos/sample (*n* = 3), using Trizol reagent (Thermo Fisher Scientific). RNA quality was evaluated using Agilent Bioanalyzer with RNA 6000 Pico kit (Agilent Technologies Japan, Ltd., Tokyo, Japan). Total low-input RNA was used for rRNA-depletion and library synthesis by Takara SMARTer kit (Takara). RNA-seq library was prepared with Agilent Bioanalyzer, DNA High-sensitivity kit (Agilent Technologies Japan, Ltd., Tokyo, Japan) and performed with Illumina NextSeq500 (Illumina K.K., Tokyo, Japan) by Tsukuba i-Laboratory LLP (Tsukuba, Japan). RNA-seq data was analyzed by CLC Genomics Workbench (Qiagen). Normalization is performed by quantile method and Log2-convert normalized value after adding one for drawing heatmap. For pairwise analysis, edgeR analysis (Empirical Analysis of DGE in CLC) was performed. ANOVA analysis (Gaussian Statistical Analysis in CLC) was performed for four groups comparison. Genes were filtrated by FDR p-value, fold change and exported in Excel format. The Enrichr program was used for GO terms and KEGG pathway enrichment analyses of differentially expressed genes with at least two-fold change and FDR < 0.05.

## Vector construction

Part of *Cables2* cDNA containing exons 1 and 2 was cloned in-frame into pBlueScript KS +at the *Bam*HI site, and the fragment containing exons 3–10 was cloned into the pcDNA3 vector at the

*Bam*HI site. These fragments were obtained and amplified from a mouse embryo E7.5 cDNA library and sequenced. The part covering *Cables2* exons 1 and 2 was cut at the *Afe*I site and ligated into the pcDNA3 vector containing exons 3–10 to synthesize the full-length *Cables2*. A 1.5 kb *Cables2* riboprobe was prepared by amplification from the full-length cDNA template with the pcDNA3 backbone, synthesized with Sp6 polymerase, and labeled with digoxigenin as a riboprobe.

A ROSA26 knock-in vector was constructed by insertion of CAG-Venus-IRES Pac gene expression cassette (*Khoa et al., 2016*) into the entry site of pROSA26-1 vector (kindly gifted from Philippe Soriano, Addgene plasmid # 21714) (*Soriano, 1999*). The *Cables2$^{d/d}$*; *ROSA$^{YFP/+}$* was generated by electroporation of the ROSA26 knock-in vector (pROSA26-CAG-Venus-IRES Pac) into *Cables2$^{d/d}$* ESCs. The CAG-tdTomato-2A and 3xFLAG sequences were inserted in the upstream and downstream of *Cables2* cDNA, respectively, to make CAG-tdTomato-2A-Cables2-3xFLAG vector. The expression of tdTomato and FLAG-tagged Cables2 in the CAG-tdTomato-2A-Cables2 vector-transfected 293T cells were evaluated by fluorescent microscopy and western blot analysis with anti-FLAG antibody, respectively (data not shown).

## Production of *Cables2* rescue chimeras by tetraploid complementation assay

Tetraploid (4n) wild-type embryos were made by electrofusing diploid (2n) embryos at two-cell-stage and cultured up to morula stage. The 4 n wild-type morula were aggregated with *Cables2$^{d/d}$*; *ROSA$^{YFP/+}$* or *Cables2$^{d/d}$*; *ROSA$^{YFP/+}$*; *CAG-tdTomato-2A-Cables2-3xFLAG* ESCs to form blastocyst chimeras. B6N wild-type ESCs was used as a control for tetraploid complementation assay. To obtain comparable control embryos at each stage of development, an equal number of control blastocyst chimeras were transferred together with *Cables2$^{d/d}$* blastocyst chimeras to a pseudopregnant recipient mouse at E2.5. Embryos were recovered at from E6.5 to E9.5 and the contribution of ESCs was evaluated by YFP or tdTomato fluorescence signals.

## Whole-mount in situ hybridization (WISH)

All embryos were dissected from the decidua in PBS with 10% fetal bovine serum and staged using morphological criteria (*Downs and Davies, 1993*) or described as the number of days of development. WISH was carried out following standard procedures, as described previously (*Rosen and Beddington, 1994*). Briefly, embryos were fixed overnight at 4°C in 4% paraformaldehyde in PBS, dehydrated, and rehydrated through a graded series of 25–50% – 75% methanol/PBS. After proteinase K (10 µg/mL) treatment for 15 min, embryos were fixed again in 0.1% glutaraldehyde/4% paraformaldehyde in PBS. Pre-hybridization at 70°C for at least 1 hr was conducted before hybridization with 1–2 µg/mL digoxigenin-labeled riboprobes at 70°C overnight. Pre-hybridization solution included 50% formamide, 4 × SSC, 1% Tween-20, heparin (50 µg/mL) (Sigma-Aldrich Japan K.K, Tokyo, Japan) and hybridization was added more yeast RNA (100 µg/mL) and Salmon Sperm DNA (100 µg/mL) (Thermo Fisher Scientific). For post-hybridization, embryos were washed with hot solutions at 70°C including 50% formamide, 4 × SSC, 1% SDS, and treated with 100 µg/mL RNase A at 37°C for 1 hr. After additional stringent hot washes at 65°C including 50% formamide, 4 × SSC, samples were washed with TBST, pre-absorbed with embryo powder, and blocked in blocking solution (10% sheep serum in TBST) for 2–5 hr at room temperature. The embryo samples were subsequently incubated with anti-digoxigenin antibody conjugated with alkaline phosphatase anti-digoxigenin-AP, Fab fragments (Roche Diagnostics K.K., Tokyo, Japan) overnight at 4°C. Extensive washing in TBST was followed by washing in NTMT and incubation in NBT/BCIP (Roche) at room temperature (RT) until color development. After completion of in situ hybridization (ISH), embryos were de-stained in PBST for 24–48 hr and post-fixed in 4% paraformaldehyde in PBS. Embryos were processed for photography through a 50%, 80%, and 100% glycerol series. Before embedding for cryosectioning, embryos were returned to PBS and again post-fixed in 4% paraformaldehyde in PBS. The specimens were placed into OCT cryoembedding solution, flash-frozen in liquid nitrogen, and cut into sections 14 µm thick using a cryostat (HM525 NX; Thermo Fisher Scientific). The following probes were used for WISH: *Bmp4* (*Jones et al., 1991*), *Brachyury (T)* (*Herrmann, 1991*), *Cer1* (*Belo et al., 1997*), *Foxa2* (*Sasaki and Hogan, 1993*), *Fgf8* (*Bachler and Neubüser, 2001*), *Lefty1/2* (*Meno et al., 1996*), *Lhx1* (*Shawlot and Behringer, 1995*), *Nanog* (*Chambers et al., 2003*), *Nodal* (*Conlon et al.,*

1994), *Oct4* (*Schöler et al., 1990*), *Otx2* (*Simeone et al., 1993*), *Sox2* (*Avilion et al., 2003*), *Sox17* (*Kanai et al., 1996*), and *Wnt3* (*Roelink et al., 1990*).

## Co-immunoprecipitation (Co-IP)

At 1 day before transfection, aliquots of $5 \times 10^4$ 293T cells were seeded onto poly-L-lysine (PLL)-coated 6 cm dishes and co-transfected with 2 µg of each pCAG-based expression vector using Lipofectamine 3000 (Thermo Fisher Scientific). After 48 hr, the cells were washed once with PBS, resuspended in RIPA buffer (50 mM Tris-HCl, pH 7.4, 150 mM NaCl, 1 mM EDTA, 1% deoxycholic acid and 1% Nonidet P-40 [NP-40]) containing protease inhibitor cocktail (Roche Diagnostics) and placed on ice for 30 min. The supernatant was collected after centrifugation and incubated with Dynabeads Protein G (Veritas Co., Tokyo, Japan) and mouse anti-FLAG antibody (F1804; Sigma-Aldrich) overnight at 4°C. The beads were washed four times with PBS, resuspended in Laemmli sample buffer, and boiled. The precipitated proteins were analysed by SDS-polyacrylamide gel electrophoresis (SDS-PAGE) and western blotting using the ECL Select Western Blotting Detection System (GE Healthcare Japan Co., Ltd., Tokyo, Japan) and a LAS-3000 imaging system (GE Healthcare). The FLAG antibody was then washed out and the membrane was re-stained with anti-β-catenin antibody (#8480, Cell Signalling Technology), anti-FLAG antibody (F1804; Sigma-Aldrich), and anti-GAPDH antibody (sc-25778, Santa Cruz).

## Luciferase reporter assay

A total of 50,000 cells were plated in PLL-coated 96-well tissue culture plates. After overnight culture, the cells were transfected with a specific promoter-driven firefly reporter plasmid and *Renilla* luciferase control plasmid, pRL-TK, using Lipofectamine 3000 Reagent (Thermo Fisher Scientific) and opti-MEM (Thermo Fisher Scientific). Luciferase activity was analysed using a luminometer and a Dual-Glo Luciferase assay kit according to the manufacturer's instructions (Promega K.K., Tokyo, Japan). The firefly luciferase values were normalized to those of *Renilla* luciferase. To evaluate β-catenin activity, cells were transiently transfected with TOPflash (TOP) or FOPflash (FOP) reporter plasmids carrying multiple copies of a wild-type or mutated TCF-binding site, respectively. Relative activity was calculated as normalized relative light units of TOPflash divided by normalized relative light units of FOPflash. Two-tailed p-values at less than 0.05 were considered as statistically significant.

## Histology, EdU, and TUNEL assay

Mouse uteri including the decidua were collected and fixed in 4% paraformaldehyde in PBS. Subsequently, paraffin blocks were made by dehydration in ethanol, clearing in xylene, and embedding in paraffin. Embryo sections 5 µm thick were cut (Microm HM 335E; Thermo Fisher Scientific) and placed on glass slides (Matsunami Glass Ind., Ltd., Osaka, Japan). For haematoxylin-eosin (HE) staining, slides were deparaffinized and rehydrated through an ethanol series, and then stained with HE.

To label the proliferating embryonic cells, pregnant mice were injected intraperitoneally with 5-ethynyl-2'-deoxyuridine (EdU) at 200 µL/mouse and sacrificed 4–6 hr later. Embryos were embedded in paraffin blocks, and sections were refixed in 4% paraformaldehyde and permeabilized in 0.5% Triton X-100/PBS. EdU assay was performed with a Click-iT Plus EdU Imaging Kit (Thermo Fisher Scientific) and TUNEL assay was performed with a Click-iT Plus TUNEL Assay for In situ Apoptosis Detection kit (Thermo Fisher Scientific) according to the manufacturer's protocol. As the final step, embryo sections were co-stained with Hoechst 33342 or DAPI, observed under a microscope (BZ-X700; Keyence). At least two sections were counted per embryo. The total stained nuclear count was assumed as total cell number, and cell number was counted using ImageJ software. The images were processed to create grayscale type, make binary and counted by the 'Analyze Particles' function in ImageJ to count the positive cells.

## Acknowledgements

We thank all members of the Sugiyama Laboratory and Laboratory Animal Resource Center for helpful discussions and encouragement. Furthermore, we are indebted to T Chiba, K Kako, H Katayama, and Y Yuda for discussion and comments on this manuscript.

## Additional information

### Funding

| Funder | Grant reference number | Author |
|---|---|---|
| Grant-in-Aid for Scientific Research(B), Japan Society for the Promotion of Science | JSPS KAKENHI 17H03568 | Fumihiro Sugiyama |
| Grant-in-Aid for Scientific Research(S), Japan Society for the Promotion of Science | JSPS KAKENHI 26221004 | Satoru Takahashi |
| Grant-in-Aid for Scientific Research(C), Japan Society for the Promotion of Science | JSPS KAKENHI 17K07130 | Hiroyoshi Iseki |
| Grant-in-Aid for Young Scientists (B), Japan Society for the Promotion of Science | JSPS KAKENHI 19K16020 | Tra Thi Huong Dinh |
| Grant-in-Aid for Scientific Research(A), Japan Society for the Promotion of Science | JSPS KAKENHI 20H00444 | Fumihiro Sugiyama |
| The Cooperative Research Project Program of Life Science Center for Survival Dynamics, Tsukuba Advanced Research Alliance (TARA Center), University of Tsukuba, Japan | 182107 | Fumihiro Sugiyama |

The funders had no role in study design, data collection and interpretation, or the decision to submit the work for publication.

### Author contributions

Tra Thi Huong Dinh, Conceptualization, Resources, Data curation, Formal analysis, Funding acquisition, Investigation, Methodology; Hiroyoshi Iseki, Conceptualization, Resources, Data curation, Formal analysis, Funding acquisition, Validation, Investigation, Methodology; Seiya Mizuno, Conceptualization, Resources, Data curation, Formal analysis, Investigation, Methodology, Project administration; Saori Iijima-Mizuno, Masatsugu Ema, Mitsuyasu Kato, Satoru Takahashi, Resources, Methodology; Yoko Tanimoto, Yoko Daitoku, Kanako Kato, Yuko Hamada, Ammar Shaker Hamed Hasan, Hayate Suzuki, Methodology; Kazuya Murata, Investigation, Methodology; Masafumi Muratani, Validation, Methodology; Jun-Dal Kim, Junji Ishida, Akiyoshi Fukamizu, Resources, Validation, Methodology; Ken-ichi Yagami, Conceptualization, Project administration; Valerie Wilson, Ruth M Arkell, Resources; Fumihiro Sugiyama, Conceptualization, Supervision, Funding acquisition, Investigation, Project administration

### Author ORCIDs

Tra Thi Huong Dinh https://orcid.org/0000-0003-1705-3865
Hiroyoshi Iseki http://orcid.org/0000-0001-5997-5564
Seiya Mizuno http://orcid.org/0000-0002-6740-5817
Yoko Tanimoto https://orcid.org/0000-0002-0731-6134
Kazuya Murata https://orcid.org/0000-0002-3328-069X
Masatsugu Ema https://orcid.org/0000-0002-6549-2611
Jun-Dal Kim https://orcid.org/0000-0002-7361-0999
Junji Ishida https://orcid.org/0000-0002-4750-6354
Akiyoshi Fukamizu https://orcid.org/0000-0002-8786-6020
Mitsuyasu Kato https://orcid.org/0000-0001-9905-2473
Satoru Takahashi https://orcid.org/0000-0002-8540-7760
Valerie Wilson http://orcid.org/0000-0003-4182-5159

Ruth M Arkell (ORCID) https://orcid.org/0000-0002-6213-7323
Fumihiro Sugiyama (ORCID) https://orcid.org/0000-0003-4744-3493

### Ethics

Animal experimentation: Animal experiments were carried out in a humane manner with approval from the Institutional Animal Experiment Committee of the University of Tsukuba (#19-057 and #20-065) in accordance with the Regulations for Animal Experiments of the University of Tsukuba and Fundamental Guidelines for Proper Conduct of Animal Experiment and Related Activities in Academic Research Institutions under the jurisdiction of the Ministry of Education, Culture, Sports, Science, and Technology of Japan.

### Decision letter and Author response

Decision letter https://doi.org/10.7554/eLife.50346.sa1
Author response https://doi.org/10.7554/eLife.50346.sa2

## Additional files

### Supplementary files

• Supplementary file 1. Differential gene expression in wild-type and *Cables2d* embryos. Statistical significance was determined using ANOVA test (FDR < 0.05). This gene list is related to heatmap in *Figure 6A*.

• Supplementary file 2. Differential gene expression in wild-type and *Cables2d* embryos at E6.5 and KEGG pathway, GO term enrichments in upregulated genes. Statistical significance was determined using EDGE test (FDR < 0.05). This gene list is related to *Figure 6B–D*.

• Supplementary file 3. Differential gene expression in wild-type and *Cables2d* embryos at E7.5 and KEGG pathway, GO term enrichments. Statistical significance was determined using EDGE test (FDR < 0.05). This gene list is related to *Figure 6E–G*.

• Supplementary file 4. CRISPR target sites for gene-modified mice generation.

• Supplementary file 5. Primers for genotyping, RT-PCR and RT-qPCR.

• Supplementary file 6. Mendelian ratio of Cables2d and Cables2e1 intercross at E9.5.

### Data availability

The RNA-seq data have been deposited in the NCBI GEO database under accession codes GSE161338.

The following dataset was generated:

| Author(s) | Year | Dataset title | Dataset URL | Database and Identifier |
|---|---|---|---|---|
| Sugiyama F, Muratani M, Thi T, Dinh H | 2020 | Comparative transcriptomic analysis between wild-type (WT) and Cables2-null embryos | https://www.ncbi.nlm.nih.gov/geo/query/acc.cgi?acc=GSE161338 | NCBI Gene Expression Omnibus, GSE161338 |

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
