## [Decision Letter]

**Acceptance summary:**

This study, which reveals an important role for the mammalian-specific Cables2 gene in embryonic development via the p53 pathway, is detailed and well-executed and brings diverse genetic and genomic evidence to bear on the mechanism of action of a locus with previously confusing knockout phenotypes.

**Decision letter after peer review:**

Thank you for submitting your article "*Cables2* is a novel Smad2-regulatory factor essential for early embryonic development in mice" for consideration by *eLife*. Your article has been reviewed by 3 peer reviewers, and the evaluation has been overseen by Michael Eisen as the Senior and Reviewing Editor. The following individual involved in review of your submission has agreed to reveal their identity: Isabelle Migeotte (Reviewer #2).

This manuscript describes the phenotype of Cables2 (Cdk5 and Abl enzyme substrate 2) mutant embryos. Cables2 is a mammal-specific protein that belongs to the Cables family whose members have a C-terminal cyclin box-like domain, and whose founding member, Cables1, is known for interacting with cyclin-dependant kinases. Cables1 deficient mice have a rather mild phenotype related to cellular proliferation in distinct organs. Cables2 deficient mutants have a much more severe phenotype as embryos arrest around E7.5 and die at mid-gestation with peri-gastrulation abnormalities.

The results are well documented, and the reviewers point to several important claims, including a completely new gene essential for early patterning, novel mechanistic insights into integration of TGF-β and Wnt signalling in the early embryo

However the reviewers feel some additional validation studies are needed, and point to an aspect of the phenotyping that was possibly misinterpreted leading to erroneous conclusions.

Specifically, the review raised the concern that the phenotype of Cables2 mutant embryos is likely the result of improper development of the conceptus brought about by defects in cell proliferation. These defects appear to have a greater effect in the epiblast, a tissue that undergoes rapid cell proliferation at pre-gastrulation and during gastrulation stages. Since gastrulation cannot proceed until a certain threshold number of cells is available in the conceptus (see Kojima et al., 2014. Sem. Cell and Dev. Biol.), this process will be delayed. Therefore, developmental events at post-implantation stages will advance at a slower rate depending on the number of cells available and in the order that they normally occur; first axial specification, then gastrulation and so on, until the embryo cannot sustain further development. Cables2 mutants appear to have undergone axial specification and begin gastrulation but then fail to develop much further. This possibility, as well as those raised below, should be addressed in the revised manuscript.

Comments on paper not needing response:

Cables2 expression is well documented and supports the author's conclusions that it is ubiquitously expressed. Although no negative controls are presented in wholemount in situ hybridization experiments in Figure 1, this data is nicely supported by expression analysis of wild-type and mutant E6.5 embryos shown in Supplementary Figure 1B.

The phenotype of the mutant embryos is clearly illustrated in Figure 2 and in the marker analysis of subsequent figures. As the authors state, mutant embryos are smaller than littermates and have peri-gastrulation abnormalities.

The mutant embryos have a smaller embryonic region that appears to represent only one third of the proximodistal length of the egg cylinder. That is, the length of the egg cylinder from the base of the ectoplacental cone to the tip of the epiblast. This is revealed by comparing the position of the boundary between the extra-embryonic ectoderm and the epiblast in mutant and control embryos (i.e. Figure 3. C, F, L and others). Mutant embryos have similar egg cylinder length as controls, but the length of the embryonic region is smaller, with a 1:2 ratio while controls have approximately a 1:1 ratio between the embryonic and extra-embryonic regions. This can be easily visualized in Figure 6F which shows a mutant embryo hybridized with Oct4, a marker of the epiblast (Perea-Gomez, et al., 2004. Curr. Biol).

Issues to address:

Cables2 mutant mice were generated using targeted ES cells obtained from KOMP repository but there is no description of the kind of mutation generated. A diagram detailing the mutation is necessary to illustrate the nature of the mutation for readers of the article.

The authors report no difference between cell proliferation and cell death levels in embryos at E6.5. However, no statistical analysis is provided to support these claims. In fact, the levels of cell proliferation, as shown in Supplementary Figure 2C, appear to differ in the embryonic region of mutant and control embryos.

Contrary to the authors' assertion, the expression of Brachyury is not normal in E6.5 mutant embryos. Previous to the appearance of the primitive streak, Brachyury is expressed as a ring in the extra-embryonic ectoderm and subsequently in the posterior epiblast (Perea-Gomez, et al., 2004. Curr. Biol; Rivera-Perez and Magnuson, 2005. Dev. Biol.). This is evident in the control embryo shown in Figure 3A. In figure 3B, Brachyury expression is clearly evident as a band in the extra-embryonic ectoderm but it is missing in the posterior epiblast. Therefore, even before the appearance of the primitive streak, Cables2 mutant embryos seem to be developmentally delayed. At E7.5, Brachyury is expressed in the primitive streak region of Cables2 mutants showing that the anteroposterior axis of mutant embryos has been properly established even though the morphology of the embryos is clearly abnormal.

Judging by morphology, the embryo hybridized with Wnt3 in Figure 3D does not appear to be a Cables2 mutant embryo. This is evident by looking at the ratio of the length of the embryonic and extra-embryonic regions. This ratio is close to 1:1 as in the control embryo shown in Figure 3C. It is likely that this is a wild-type or heterozygous embryo that is delayed in development rather than a Cables2 mutant embryo.

The authors state: "….expression of Wnt3 was also impaired in the proximal-posterior part of epiblast and the PVE of E6.5 Cables2-/- mutants although the expression remained in the proximal epiblast adjacent to the extraembryonic ectoderm (ExE)…." Apart from the previous comment, no evidence is provided to support this claim. A cross section of Cables2 mutant embryos hybridized with Wnt3 in necessary to support this claim.

The authors state: "WISH analyses showed that Bmp4 was similarly expressed in the ExE of Cables2-/- embryos compared with wild-type embryos at E6.5 (Figure 3K and L), suggesting that the ExE is normally developed in mutant embryos at least until E6.5. These findings are consistent with Cables2 promoting the formation of Wnt3-expressing PVE to induce and maintain PS formation." There appears to be a contradiction, if expression of Bmp4 and extra-embryonic ectoderm are normal, why do the authors mention that Wnt3 expression is impaired?

The authors conclude that Cables2 depletion impairs the correct formation of the AVE (Anterior Visceral Endoderm). This is incorrect, expression of Lefty2 and Cer1, two markers of the AVE, is located on one side of the epiblast indicating correct positioning of the AVE. The small/weak area of expression described can be explained by the small embryonic region as described in point 5 above. In addition, the correct position of Fgf8 and later Brachyury indicate correct positioning of the anteroposterior axis of the Cables2 mutant embryos.

The chimera analysis in which Cables2 mutant ES cells were aggregated with wild-type tetraploid embryos shows only partial rescue of the mutant chimeric embryos suggesting a role for Cables2 in visceral endoderm for embryo development but not an essential requirement in this tissue. If this were the case, the mutant chimeras should have been rescued to stages when the function of visceral endoderm is no longer required for embryo development, likely past placental stages when this organ takes over the function of visceral endoderm in the yolk sac. The results of chimeras in which Cables2 is ectopically expressed in ES cells does not provide evidence about the requirement of Cables2 in the epiblast, since in these chimeras the visceral endoderm layer is wild type. These chimeral only provide evidence of rescue of the function of Cables2 in the epiblast. The proper experiment would be to generate chimeras containing wild-type epiblast and mutant visceral endoderm.

As embryos are slightly smaller from E6.5, and then really abnormal at E7.5, it seems like a possible impact on proliferation should be explored at later time points than E6.5, particularly as the role of Cables2 on embryo growth is not elucidated. Several approach, including Brdu, Phh3 staining, or Fucci reporters might help reveal a possible role on proliferation. Should this be the case, it would be interesting to explore Cables2 interaction with cyclin-dependant kinases in the embryo.

Mutant embryos are smaller, but evidently neither cell proliferation nor apoptosis are significantly different. In my view the EdU and TUNEL experiments are not sufficiently powered and in any case are executed weakly. Samples were sectioned, and "At least 2 slides were counted per embryo". It is not clear how many sections/slide and how representative. The authors should have looked at more embryos and attempted to make measurements within the entire embryo (or at least, all the sections for a particular embryo). Moreover, the statistical test could be more robust, something like a nested Anova rather than T-test, because sections (and division/apoptotic events) from the same embryo are not independent. One suggestion from the reviewers is to remove these data, as the precise reason why the embryos are smaller is a relatively secondary point which, as they say in the discussion, need to be further elucidated.

The RNASeq data seems underused. Even if the raw data is available, it would be better to have a more detailed analysis than one limited to GO terms.

Similarly, as the authors have generated EpiLCs and propose a role for Cables2 in major pathways involved in germ layers specification, it could be interesting to explore the differentiation potential of mutant cells.

[Editors' note: further revisions were suggested prior to acceptance, as described below.]

Thank you for resubmitting your work entitled "Disruption of entire *Cables2* locus leads to embryonic lethality by diminished *Rps21* expression and enhanced p53 pathway" for further consideration by *eLife*. Your revised article has been reviewed by 3 peer reviewers and the evaluation has been overseen by Michael Eisen as the Senior and Reviewing Editor.

There is consensus among the reviewers that this is a significantly improved manuscript, and there is support for its publication. However, with the addition of so much more data, some additional concerns and questions have come up, and we hope you will be able to address these with some additional changes to the manuscript. I have elected to pass on the individual comments rather than combine them into a single review, as I think it will be easier for you to address them.

Reviewer #1:

This manuscript focuses on a mutant allele of the Cables2 gene obtained from the Knockout Mouse Consortium. The mutation is a large deletion of 13.4 Kb that removes the entire coding sequence of the gene. Homozygous mutants are embryonic lethal and have a gastrulation phenotype that is characterized by delayed development and a reduced epiblast region of the conceptus. Expression profiling revealed up-regulation of p53 that coincides with increased apoptosis in the embryos. A second mutation in Cables2 that deletes exon1 and is likely a null allele, however, leads to viable and fertile offspring. This contradicts the conclusions of the authors that attribute the gastrulation phenotype to absence of Cables2.

Expression profiling revealed down regulation of Rps21, a gene abutting the Cables2 locus. Therefore, an alternative explanation is that the large deletion generated in the Cables2 locus leads to misregulation of Rps21. One possibility is the removal of a regulatory region that controls Rps21 expression.

The authors have generated a mutation of Rps21 using CRISPR-Cas9, however, it appears that only founder mice were assessed in their study. This complicates the analysis of the mutation since founders are likely mosaics composed of wild-type cells and cells carrying different mutations in Rps21. The authors also conducted Cables2 epiblast rescue experiments using chimeras generated by aggregation of tetraploid embryos and Cables2 mutant ES cells. These studies are ambiguous since they do not distinguish between rescue of epiblast or of extra-embryonic tissues.

In summary, there are multiple caveats that cast doubt in the conclusions of the article. The generation of compound heterozygous between the two Cables2 null alleles and careful analysis of inactivating mutations in Rps21 should help differentiate the roles played by Cables2 and Rps21 genes in gastrulation. Determining if transcripts of Cables2 are present in Cables2 exon1 mutants during gastrulation can also help clarify the results.

The authors have addressed my previous critiques satisfactorily. However, a significant amount of new data has been added to the original manuscript. Although this is commendable, it has led to new challenges to the original conclusions of the manuscript.

1. The authors have provided straightforward whole-mount in situ hybridization (WISH) data of E6.5 of mutant embryos that clearly illustrates the phenotype of the original Cables2 mutant allele (Cables2d herein). However, WISH data for Oct4 and Nodal that highlight the reduced size of the epiblast in Cables2d mutants has been removed. This is essential data for understanding the phenotype that should be reinstated.

2. The cell proliferation and apoptosis assays at E6.5 have been replaced by analyses at E7.5. This was done to illustrate the high levels of apoptosis in Cables2d mutants that were not evident at E6.5. This is not a proper comparison since the Cables2d mutants are clearly less developed than E7.5 controls. Showing cell proliferation and apoptosis assays at E6.5 and then showing the increase in apoptosis as the mutant embryos progress in development is a better way to illustrate the point.

3. The authors need to stress the presence of primitive streak markers in the posterior epiblast at E6.5 (before morphological appearance of the primitive streak) and establishment of the anteroposterior axis rather than focus on the low levels of expression (or absence of expression in the case of T).

4. Expression profiling of Cables2d mutants showed that Rps21 expression is decreased. This is very intriguing since the deletion of Cables2 encompasses 13.4 kb and it is located right next to Rps21. It is possible that this deletion removed a regulatory region that affects the expression of Rps21 rather than indicate down-regulation of the gene caused by the absence of Cables2. The absence of phenotype in a Cables2 allele that removes exon 1 (Cables2d1 herein), a null allele as indicated by the authors, further suggest that the removal of Cables2 is not responsible for the phenotype observed in Cables2d mutants. Compound heterozygous containing the two mutant alleles of Cables2 (Cables2d/Cables2d1) can help address this possibility.

5. The authors used the CRISPR-Cas9 system to generate a null mutation of Rps21 that removes exons 2-6. They report the generation of heterozygous animals with a semidominant lethal or sterile phenotype characterized by white belly spotting, kinky tail and small size. However, it appears that the analysis was restricted to founder animals. Founders generated using the CRISPR-Cas9 system are generally mosaic animals. Thus, they can be composed of an amalgam of heterozygous, homozygous and wt cells. Also, they may contain a variety of indels apart from the intended mutation. Therefore, these results need to be re-evaluated after germline transmission and confirmation of the genotype. In addition, the phenotype may be compounded by defects in a miRNA gene (Mir3091, as per ensembl genome browser) located in intron 1 of Rps21.

6. The authors generated chimeras using mutant Cables2 ES cells that carry fluorescent markers and express Cables2 under the CAG promoter. These chimeras provide evidence of rescue of the Cables2 mutation in the epiblast. However, these chimeras also carry tetraploid wt extra-embryonic tissues and the authors showed that these tissues also rescue the phenotype in chimeras. Therefore, it is not clear if the rescue is due to rescue of Cables2 in the epiblast or rescue of Cables2 function in extra-embryonic tissues.

7. Lines 149-152. "Prior to gastrulation, T transcripts are first detected in a ring at the embryonic/extraembryonic junction, whereas once gastrulation is initiated they are found in the PS and nascent mesoderm and subsequently in the axial mesendoderm (Wilkinson et al., 1990)."

T expression is first detected as a ring in extra-embryonic ectoderm and then in the posterior epiblast before the appearance of the primitive streak. Please correct this paragraph and add the following references (Perez-Gomez et al., Curr. Biol. 2004; Rivera-Perez and Magnuson, Dev. Biol. 2005).

8. Lines 160-161. The following references that report Wnt3 expression in posterior epiblast and PVE need to be added: Liu et al., Nat. Gen. 2004 and Rivera-Perez and Magnuson, Dev. Biol. 2005.

9. Lines 161-163. "WISH showed that expression of Wnt3 appeared mostly in the proximal epiblast…"

WISH does not prove that Wnt3 is expressed in the proximal epiblast. To conclusively demonstrate expression in the epiblast, the Wnt3-hybridized embryos need to be cross sectioned.

10. Line 164-165. "The ExE of the post-implantation mouse embryo expresses Bmp4 whereby promoting Wnt3 expression in the proximal epiblast for PS formation."

Expression of Bmp4 in extra-embryonic ectoderm does not prove that Bmp4 promotes expression of Wnt3 in proximal epiblast for PS formation.

11. Lines 168-169. "These findings are consistent with Cables2 promoting the formation of Wnt3-expressing PVE to induce and maintain PS formation in the Bmp4-independant manner"

This conclusion overstretches the results.

12. Lines 178-180. "The combined results of WISH analyses suggested the AVE and PS formation are retarded and impaired at the onset of gastrulation in the Cables2-null model."

I agree that PS formation is delayed but no evidence is provided to show that AVE formation is delayed too. Certainly, markers of AVE are expressed weakly but they are present at the right place at the right time.

Reviewer #2:

The manuscript is quite different from the first version, as additional inquiry has shown an important role for the diminished expression of Rps21, a gene partially located on the cables2 locus in the opposite orientation. The authors provide a detailed description of the Cables2 locus mutant, the Rps21 mutant, and the Cables2 only mutant, as well as dissect Cables2 roles in distinct embryo germ layers.

The study is interesting and technically very well executed. The authors have addressed the previous review's requests.

However, since the additional experiments led to such a change of message, I feel the current narrative of the manuscript a bit confusing. It would seem easier for the readers to re-write the story as a clear dissection of the interaction between the defects related to ablating one or the other genes or both. As the first half was not modified, the reader is confronted to a complete description of the Cables2 phenotype before realising the genetic complexity.

Reviewer #3:

This manuscript examines the role of Cables2 during early mammalian development. The authors find profound peri-gastrulation defects, that they unpick at reasonable depth.

What is interesting:

1. A completely new gene essential for early patterning.

2. Novel mechanistic insights into integration of TGF-β and Wnt signalling in the early embryo.

Main concern:

1. Figure 6H: The authors have not really addressed my previous concern regarding the validity of the quantitation of cell proliferation/death. Moreover, the detail they have provided in the methods is inadequate.

I copy here my concerns from the previous review, that remain unaddressed "It is not clear how many sections/slide and how representative. The authors should have looked at more embryos and attempted to make measurements within the entire embryo (or at least, all the sections for a particular embryo). Moreover, the statistical test could be more robust, something like a nested Anova rather than T-test, because sections (and division/apoptotic events) from the same embryo are not independent."

Lines 624-625: Need to give more details on how cell number (I assume it was nuclear number) was counted – what plug-in, what parameters etc.?

Figure 6A – G: need to provide scale bars.

---

## [Author Response]

[…] Issues to address:Cables2 mutant mice were generated using targeted ES cells obtained from KOMP repository but there is no description of the kind of mutation generated. A diagram detailing the mutation is necessary to illustrate the nature of the mutation for readers of the article.

Thank you very much for reminding us. We added the description of the Velocigene knockout strategy in Results, line 123-125. Furthermore, the targeting vector design is also shown in Figure 2—figure supplement 1A.

The authors report no difference between cell proliferation and cell death levels in embryos at E6.5. However, no statistical analysis is provided to support these claims. In fact, the levels of cell proliferation, as shown in Supplementary Figure 2C, appear to differ in the embryonic region of mutant and control embryos.

Thank you for consideration. Previously, we described the statistical analysis Student t-test in the figure legend of embryo E6.5. However, the new data from E7.5 embryos showed significantly increasing apoptosis in Cables2-null embryos. Therefore, we would like to show only data of E7.5 embryo in Figure 6.

Contrary to the authors' assertion, the expression of Brachyury is not normal in E6.5 mutant embryos. Previous to the appearance of the primitive streak, Brachyury is expressed as a ring in the extra-embryonic ectoderm and subsequently in the posterior epiblast (Perea-Gomez, et al., 2004. Curr. Biol; Rivera-Perez and Magnuson, 2005. Dev. Biol.). This is evident in the control embryo shown in Figure 3A. In figure 3B, Brachyury expression is clearly evident as a band in the extra-embryonic ectoderm but it is missing in the posterior epiblast. Therefore, even before the appearance of the primitive streak, Cables2 mutant embryos seem to be developmentally delayed. At E7.5, Brachyury is expressed in the primitive streak region of Cables2 mutants showing that the anteroposterior axis of mutant embryos has been properly established even though the morphology of the embryos is clearly abnormal.

Thank you very much for pointing out. We modified the description of *T* in line 153-154 following your comment. In agreement with you, we described that the signal is exhibited, but decreased intensity relative to wild-type embryos. This result is consistent with the decreased *Wnt3* and *Fgf8* expression in the posterior part. We thought that *T* expression is normal until E6.0, therefore the primitive streak formation was started and initiated, however, right after, the embryo failed to maintain the gastrulation. This phenotype was also considered in *Wnt3*-deficient VE/epiblast embryo (Tortelote, et al., 2013. Dev. Biol., Yoon et al., 2015. Dev. Biol.).

Judging by morphology, the embryo hybridized with Wnt3 in Figure 3D does not appear to be a Cables2 mutant embryo. This is evident by looking at the ratio of the length of the embryonic and extra-embryonic regions. This ratio is close to 1:1 as in the control embryo shown in Figure 3C. It is likely that this is a wild-type or heterozygous embryo that is delayed in development rather than a Cables2 mutant embryo.

We appreciated your comment and changed figure of *Wnt3* in Figure 3D.

The authors state: "….expression of Wnt3 was also impaired in the proximal-posterior part of epiblast and the PVE of E6.5 Cables2-/- mutants although the expression remained in the proximal epiblast adjacent to the extraembryonic ectoderm (ExE)…." Apart from the previous comment, no evidence is provided to support this claim. A cross section of Cables2 mutant embryos hybridized with Wnt3 in necessary to support this claim.

Following your requirement, we changed the description of *Wnt3* in line 161-162: “…expression of *Wnt3* appeared mostly in the proximal epiblast adjacent to the extraembryonic ectoderm (ExE) and decreased in the posterior part of E6.5 *Cables2^-/-^* mutants”. The new RNA-seq data clearly showed the downregulation of Wnt signaling pathway in Cables2-null embryos, therefore we discussed about the diminished Wnt/βcatenin signaling involved in retardation of AVE and PS formation in Discussion part, line 378-393.

The authors state: "WISH analyses showed that Bmp4 was similarly expressed in the ExE of Cables2-/- embryos compared with wild-type embryos at E6.5 (Figure 3K and L), suggesting that the ExE is normally developed in mutant embryos at least until E6.5. These findings are consistent with Cables2 promoting the formation of Wnt3-expressing PVE to induce and maintain PS formation." There appears to be a contradiction, if expression of Bmp4 and extra-embryonic ectoderm are normal, why do the authors mention that Wnt3 expression is impaired?

Thank you for your comment. We would like to clarify this statement clearly. Because *Bmp4* is known to be expressed in the ExE of the post-implantation mouse embryo where it promotes *Wnt3* expression in the proximal epiblast for PS formation. So the result of *Bmp4* and the extra-embryonic ectoderm is normal, while *Wnt3* is decreased, indicate that *Cables2* may promoting the *Wnt3* in the *Bmp4*-independant manner. We also added this sentence to manuscript line 168.

The authors conclude that Cables2 depletion impairs the correct formation of the AVE (Anterior Visceral Endoderm). This is incorrect, expression of Lefty2 and Cer1, two markers of the AVE, is located on one side of the epiblast indicating correct positioning of the AVE. The small/weak area of expression described can be explained by the small embryonic region as described in point 5 above. In addition, the correct position of Fgf8 and later Brachyury indicate correct positioning of the anteroposterior axis of the Cables2 mutant embryos.

Thank you very much for your comment. We are sorry that the “impair” word lead to misunderstanding. In agreement with you, we though that formation of AVE was delayed and decreased in Cables2 deletion, but not the mislocation or incorrect position. We modified that statement in line 175 and 179.

The chimera analysis in which Cables2 mutant ES cells were aggregated with wild-type tetraploid embryos shows only partial rescue of the mutant chimeric embryos suggesting a role for Cables2 in visceral endoderm for embryo development but not an essential requirement in this tissue. If this were the case, the mutant chimeras should have been rescued to stages when the function of visceral endoderm is no longer required for embryo development, likely past placental stages when this organ takes over the function of visceral endoderm in the yolk sac. The results of chimeras in which Cables2 is ectopically expressed in ES cells does not provide evidence about the requirement of Cables2 in the epiblast, since in these chimeras the visceral endoderm layer is wild type. These chimeral only provide evidence of rescue of the function of Cables2 in the epiblast. The proper experiment would be to generate chimeras containing wild-type epiblast and mutant visceral endoderm.

We are very sorry for confusing about reviewer’s comment: “The results of chimeras in which Cables2 is ectopically expressed in ES cells does not provide evidence about the requirement of Cables2 in the epiblast (visceral endoderm?), since in these chimeras the visceral endoderm layer is wild type.”

Our chimera analysis does not provide strong evidence about the essential requirement of *Cables2* in the visceral endoderm, but suggesting a role for *Cables2* in visceral endoderm and the requirement of *Cables2/Rps21* in the epiblast. Previously, the rescued phenotype in tetraploid embryos by overexpression of *Cables2* strongly indicated the requirement of *Cables2* in epiblast. However, since we found the decreased *Rps21* expression and increased apoptosis in *Cables2*-null embryo, the rescued phenotype became more important to suggest the involvement of Cables2 to embryogenesis via unknown mechanism.

As embryos are slightly smaller from E6.5, and then really abnormal at E7.5, it seems like a possible impact on proliferation should be explored at later time points than E6.5, particularly as the role of Cables2 on embryo growth is not elucidated. Several approach, including Brdu, Phh3 staining, or Fucci reporters might help reveal a possible role on proliferation. Should this be the case, it would be interesting to explore Cables2 interaction with cyclin-dependant kinases in the embryo.

We appreciated and totally agreed with reviewer about the analysis in later stage of embryo. The EdU and TUNEL assay using embryo E7.5 indicated that not the proliferation but the apoptosis is the main cause of lethal phenotype in Cables2-null embryo. For your more information, Cdk5, which bind to C-ter of Cables2, also showed the abnormal phenotype and died after E16.5, later stage than Cables2-KO (Ohshima et al. 1996. PNAS).

Furthermore, except Cdk1 and Cyclin A2-deficient mice, other KO mice of Cdk and Cyclin are viable or died in later stage than Cables2 (reviewed by Satyanarayana and Kaldis, 2009. Oncogene). Therefore, we expected that Cables2 has an essential function for mouse embryogenesis and reveal its novel interaction with Rps21, p53 and Wnt/β-catenin signaling in this manuscript.

Mutant embryos are smaller, but evidently neither cell proliferation nor apoptosis are significantly different. In my view the EdU and TUNEL experiments are not sufficiently powered and in any case are executed weakly. Samples were sectioned, and "At least 2 slides were counted per embryo". It is not clear how many sections/slide and how representative. The authors should have looked at more embryos and attempted to make measurements within the entire embryo (or at least, all the sections for a particular embryo). Moreover, the statistical test could be more robust, something like a nested Anova rather than T-test, because sections (and division/apoptotic events) from the same embryo are not independent. One suggestion from the reviewers is to remove these data, as the precise reason why the embryos are smaller is a relatively secondary point which, as they say in the discussion, need to be further elucidated.

We highly appreciated the comment and suggestion of reviewer. So far, we do not show the data of E6.5 and show only data of E7.5. The TUNEL-positive cells are significantly increased in mutant embryo as shown in Figure 6. The RNA-seq data further supported that elevated p53 pathway caused the apoptosis in Cables2 mutant embryo. However, the unknown mechanism should be further elucidated in the future studies.

The physical interaction of Cables2 with Smad2 and β-catenin could be verified within the embryo, either by coIP or proximity ligation assay (which would also provide spatial information).

We thank so much for this suggestion. However, with the limitation in collecting protein from embryo samples at E6.5~ E8.5, we have not performed the CoIP yet. Besides, we did contact Sigma-Aldrich for buying the kit of Proximity ligation assay last year. However, that kit could not be imported to Japan so far. Recently, our team generated a Cables2 reporter mice. With high amount of protein (1 ~ 2 µg) collecting from this mice, we showed the interaction of Cables2 with Cdk5 in testis and brain of adult mice by CoIP (Hasan et al. 2020. Exp. Anim.). Therefore, up to date, we could not perform the CoIP of Cables2 with other factors using embryo samples.

The RNASeq data seems underused. Even if the raw data is available, it would be better to have a more detailed analysis than one limited to GO terms.

Thank you very much for this comment. In this revised manuscript, we added the new RNAseq data of embryo, not the EpiLC.

Similarly, as the authors have generated EpiLCs and propose a role for Cables2 in major pathways involved in germ layers specification, it could be interesting to explore the differentiation potential of mutant cells.

We appreciated for reviewer’s suggestion. Recently, we found that Cables2 functions together with Rps21, p53 and Wnt signaling. Therefore, we want to explore firstly the mechanism and specific function of Cables2 with those factors. But we thank reviewer so much for your nice recommendation.

Finally, the consequence of down-regulation of Nanog expression in mutants could be explored in more details, including possibly through connecting with the RNAseq data.

We are very sorry for excluding Nanog data in this revised manuscript. Since the newest data did not completely support our hypothesis about Nodal/Smad2 in gastrulation, we just withdrawn the Nodal, Smad2 and Nanog data from this manuscript. Anyway, we really appreciated suggestion from reviewer.

[Editors' note: further revisions were suggested prior to acceptance, as described below.]

Reviewer #1:[…] The authors have addressed my previous critiques satisfactorily. However, a significant amount of new data has been added to the original manuscript. Although this is commendable, it has led to new challenges to the original conclusions of the manuscript.1. The authors have provided straightforward whole-mount in situ hybridization (WISH) data of E6.5 of mutant embryos that clearly illustrates the phenotype of the original Cables2 mutant allele (Cables2d herein). However, WISH data for Oct4 and Nodal that highlight the reduced size of the epiblast in Cables2d mutants has been removed. This is essential data for understanding the phenotype that should be reinstated.

Thank you very much for comment, we reinstated the data of *Oct4* and *Nodal* in Figure 3I-L.

2. The cell proliferation and apoptosis assays at E6.5 have been replaced by analyses at E7.5. This was done to illustrate the high levels of apoptosis in Cables2d mutants that were not evident at E6.5. This is not a proper comparison since the Cables2d mutants are clearly less developed than E7.5 controls. Showing cell proliferation and apoptosis assays at E6.5 and then showing the increase in apoptosis as the mutant embryos progress in development is a better way to illustrate the point.

As reviewer’s requirement, we reinstated the cell proliferation and apoptosis data at E6.5 in Figure 5. For the convenience and consistency, the data of E7.5 were also combined into Figure 5.

3. The authors need to stress the presence of primitive streak markers in the posterior epiblast at E6.5 (before morphological appearance of the primitive streak) and establishment of the anteroposterior axis rather than focus on the low levels of expression (or absence of expression in the case of T).

We appreciate your comment and describe more about primitive streak markers in line 150-157 and line 180-182.

4. Expression profiling of Cables2d mutants showed that Rps21 expression is decreased. This is very intriguing since the deletion of Cables2 encompasses 13.4 kb and it is located right next to Rps21. It is possible that this deletion removed a regulatory region that affects the expression of Rps21 rather than indicate down-regulation of the gene caused by the absence of Cables2. The absence of phenotype in a Cables2 allele that removes exon 1 (Cables2d1 herein), a null allele as indicated by the authors, further suggest that the removal of Cables2 is not responsible for the phenotype observed in Cables2d mutants. Compound heterozygous containing the two mutant alleles of Cables2 (Cables2d/Cables2d1) can help address this possibility.

Thank you very much for this recommendation. In accordance with the comment, we intercrossed and analyzed the compound heterozygous at E9.5, as shown in Supplementary Table 1 and Figure 7—figure supplementary 1. We described about *Cables2^d/e1^* compound mice in the Discussion. The survival of the compound gastrulas explained the importance of the threshold of *Rps21* expression for embryonic development.

5. The authors used the CRISPR-Cas9 system to generate a null mutation of Rps21 that removes exons 2-6. They report the generation of heterozygous animals with a semidominant lethal or sterile phenotype characterized by white belly spotting, kinky tail and small size. However, it appears that the analysis was restricted to founder animals. Founders generated using the CRISPR-Cas9 system are generally mosaic animals. Thus, they can be composed of an amalgam of heterozygous, homozygous and wt cells. Also, they may contain a variety of indels apart from the intended mutation. Therefore, these results need to be re-evaluated after germline transmission and confirmation of the genotype. In addition, the phenotype may be compounded by defects in a miRNA gene (Mir3091, as per ensembl genome browser) located in intron 1 of Rps21.

Thank reviewer for this comment. We indeed confirmed the germline transmission and *Bst* phenotype in F1 mice. However, there is difficult for mouse colony maintenance in this strain, because of extremely low fertility. In this study and manuscript, we focus to *Cables2* gene locus and its mutant phenotype. Therefore, we would like not show the Rps21 phenotype data but described the F1 phenotype in line 282-283. In null *Rps21*, we designed to knock-out *Rps21* only and refrain to disrupt *Mir3091*. There is no report of *Mir3091* KO mice so far. Furthermore, there was no significantly change of *Mir3091* expression by RNAseq analysis in *Cables2d* embryos.

6. The authors generated chimeras using mutant Cables2 ES cells that carry fluorescent markers and express Cables2 under the CAG promoter. These chimeras provide evidence of rescue of the Cables2 mutation in the epiblast. However, these chimeras also carry tetraploid wt extra-embryonic tissues and the authors showed that these tissues also rescue the phenotype in chimeras. Therefore, it is not clear if the rescue is due to rescue of Cables2 in the epiblast or rescue of Cables2 function in extra-embryonic tissues.

We appreciated for this comment. Our data of chimeras confirmed the lethal phenotype of *Cables2d* mutant embryo and the important of *Cables2* locus exclusively in epiblast. Therefore, we modified the name of strategy in this revised manuscript. The name of “*Cables2* VE rescue chimera” was changed to “*Cables2d* Epi KO chimera”. Generally, this result provides evidence of overexpressed Cables2 in the epiblast can rescue the lethal phenotype of *Cables2d*.

7. Lines 149-152. "Prior to gastrulation, T transcripts are first detected in a ring at the embryonic/extraembryonic junction, whereas once gastrulation is initiated they are found in the PS and nascent mesoderm and subsequently in the axial mesendoderm (Wilkinson et al., 1990)."T expression is first detected as a ring in extra-embryonic ectoderm and then in the posterior epiblast before the appearance of the primitive streak. Please correct this paragraph and add the following references (Perez-Gomez et al., Curr. Biol. 2004; Rivera-Perez and Magnuson, Dev. Biol. 2005).

We added the references and corrected paragraph as reviewer’ suggestion.

8. Lines 160-161. The following references that report Wnt3 expression in posterior epiblast and PVE need to be added: Liu et al., Nat. Gen. 2004 and Rivera-Perez and Magnuson, Dev. Biol. 2005.

We thank a lot and added the references.

9. Lines 161-163. "WISH showed that expression of Wnt3 appeared mostly in the proximal epiblast…"WISH does not prove that Wnt3 is expressed in the proximal epiblast. To conclusively demonstrate expression in the epiblast, the Wnt3-hybridized embryos need to be cross sectioned.

Thank you very much for this recommendation. We modified the sentence and description of *Wnt3* in line 162 to focus to the appearance of *Wnt3* in mutant embryo.

10. Line 164-165. "The ExE of the post-implantation mouse embryo expresses Bmp4 whereby promoting Wnt3 expression in the proximal epiblast for PS formation."Expression of Bmp4 in extra-embryonic ectoderm does not prove that Bmp4 promotes expression of Wnt3 in proximal epiblast for PS formation.

We appreciated for this correction and modified the sentence in line 165.

11. Lines 168-169. "These findings are consistent with Cables2 promoting the formation of Wnt3-expressing PVE to induce and maintain PS formation in the Bmp4-independant manner"This conclusion overstretches the results.

We deleted this description.

12. Lines 178-180. "The combined results of WISH analyses suggested the AVE and PS formation are retarded and impaired at the onset of gastrulation in the Cables2-null model."I agree that PS formation is delayed but no evidence is provided to show that AVE formation is delayed too. Certainly, markers of AVE are expressed weakly but they are present at the right place at the right time.

In agreement with reviewer, we modified the conclusion and focused to phenotype of PS formation.

Reviewer #2:The manuscript is quite different from the first version, as additional inquiry has shown an important role for the diminished expression of Rps21, a gene partially located on the cables2 locus in the opposite orientation. The authors provide a detailed description of the Cables2 locus mutant, the Rps21 mutant, and the Cables2 only mutant, as well as dissect Cables2 roles in distinct embryo germ layers.The study is interesting and technically very well executed. The authors have addressed the previous review's requests.However, since the additional experiments led to such a change of message, I feel the current narrative of the manuscript a bit confusing. It would seem easier for the readers to re-write the story as a clear dissection of the interaction between the defects related to ablating one or the other genes or both. As the first half was not modified, the reader is confronted to a complete description of the Cables2 phenotype before realising the genetic complexity.

We appreciated reviewer #2 for your kind comments and we are glad to hear that we could address all your requests. If re-writing all the manuscript, we are not sure that we would be able to fully satisfy other reviewers and editors. However, from your kindly suggestion, we re-wrote the results of *Rps21* KO (*Rps21d*) mouse and *Cables2* exon1 deletion mice (*Cables2e1*). In addition, we gave more information in all figures of gene construction (Figure 2A, 7A, 8A), which indicated *Rps21* abutting to *Cables2* locus. Furthermore, if our manuscript meets all the criteria to be published, e*Life* digest part will be definitely helpful for explaining briefly our story.

Reviewer #3:[…] 1. Figure 6H: The authors have not really addressed my previous concern regarding the validity of the quantitation of cell proliferation/death. Moreover, the detail they have provided in the methods is inadequate.I copy here my concerns from the previous review, that remain unaddressed "It is not clear how many sections/slide and how representative. The authors should have looked at more embryos and attempted to make measurements within the entire embryo (or at least, all the sections for a particular embryo). Moreover, the statistical test could be more robust, something like a nested Anova rather than T-test, because sections (and division/apoptotic events) from the same embryo are not independent."

We appreciate reviewer #3 for indicating new finding of our manuscript. About the histology, basically, 2-3 sections were embed per slide and 2-3 slides containing the biggest vertical sections were selected and stained. We avoided the adjacent sections as trying to report independent observations. We also used the 3-4 slides for genotyping embryos, therefore we could not measure the entire embryos. Instead, we increased the number of embryos in each group, n=6 at E6.5 and n=7 at E7.5. As your recommendation, we performed two-way ANOVA statistical test and combined data of both E6.5 and E7.5.

Lines 624-625: Need to give more details on how cell number (I assume it was nuclear number) was counted – what plug-in, what parameters etc.?

Thank you for this suggestion. Total number of cells was counted by nuclear number as reviewer assumed. We added more description in Method part.

Figure 6A – G: need to provide scale bars.

Thank you so much for reminding, we added the scale bars.